# FGFR2 is essential for salivary gland duct homeostasis and MAPK-dependent seromucous acinar cell differentiation

Marit H. Aure [1] ✉, Jennifer M. Symonds[1], Carlos U. Villapudua[1], Joshua T. Dodge [1], Sabine Werner [2], Wendy M. Knosp [1] & Matthew P. Hoffman [1] ✉

Exocrine acinar cells in salivary glands (SG) are critical for oral health and loss of functional acinar cells is a major clinical challenge. Fibroblast growth factor receptors (FGFR) are essential for early development of multiple organs, including SG. However, the role of FGFR signaling in specific populations later in development and during acinar differentiation are unknown. Here, we use scRNAseq and conditional deletion of murine FGFRs in vivo to identify essential roles for FGFRs in craniofacial, early SG development and progenitor function during duct homeostasis. Importantly, we also discover that FGFR2 via MAPK signaling is critical for seromucous acinar differentiation and secretory gene expression, while FGFR1 is dispensable. We show that FGF7, expressed by myoepithelial cells (MEC), activates the FGFR2-dependent seromucous transcriptional program. Here, we propose a model where MEC-derived FGF7 drives seromucous acinar differentiation, providing a rationale for targeting FGFR2 signaling in regenerative therapies to restore acinar function.

Exocrine glands secrete fluids essential for maintenance of their target tissues[1]. Exocrine salivary glands (SG) are critical for oral health as they generate saliva for mastication, oral microbiome maintenance, and lubrication of the oral cavity. There are three major mammalian SGs, submandibular (SMG), sublingual (SLG), and parotid (PG) as well as minor salivary glands (MSG), which all differ in the composition of saliva they secrete[2,3]. The bulk of saliva secretion comes from highly specialized acinar cells that are defined as either seromucous, mucous or serous depending on the protein contents of the saliva. Despite the central role of acinar cells, little is known about the developmental mechanisms that drive their specific secretory profiles, although Kras activation was recently shown to promote acinar fate[4]. In vivo, acinar cells are surrounded by myoepithelial cells (MEC) that wrap around and directly contact them, and acini are connected to ducts, containing basal cells and ionocytes, that modify and transport saliva into the

oral cavity. Loss of acinar cells is a common feature of pathologies including autoimmune diseases such as Sjögren's disease and a side effect of irradiation therapy for head and neck cancer. Understanding acinar regeneration for the restoration of salivary function continues to be a major clinical challenge[5–7].

Preclinical studies have shown through lineage tracing that acinar and MECs are self-maintained, while basal duct cells are restricted progenitors during homeostasis[6,8]. Following injury, all cell compartments have regenerative potential, however, the contribution from any lineage is dependent on the degree and type of injury[6]. This has resulted in an updated view of stemness and plasticity in SGs leading to a renewed focus on niche signals and microenvironments that may inform regenerative therapies[9]. Identifying targets that could be used to regenerate salivary function using either molecular, genetic or cell-based approaches is an important therapeutic aim.

[1]Matrix and Morphogenesis Section, National Institute of Dental and Craniofacial Research, National Institutes of Health, Bethesda, MD, USA. [2]Institute of Molecular Health Sciences, Department of Biology, Swiss Federal Institute of Technology (ETH), Zurich, Zurich, Switzerland. ✉e-mail: marit.aure@nih.gov; mhoffman@nih.gov

Fibroblast growth factor receptors (FGFR) are a family of four receptor tyrosine kinases (FGFR1-4). Alternative splicing of an extra-cellular Ig-like III domain results in seven functionally distinct receptors expressed in a tissue specific manner; mesenchymal cells express the "c" isoforms, while epithelial cells express the "b" isoforms[10]. The epithelial FGFR1b and FGFR2b are activated in an autocrine or paracrine manner by their major ligands FGF7 and FGF10 and require heparan sulfate coreceptors for optimal signaling[11,12]. FGFR signaling is involved in development, progenitor cell proliferation, and tumorigenesis of multiple organ systems including SGs[10,12]. The critical role for FGFR signaling in human SGs is evident by haploinsufficiency of either FGFR2b or FGF10, which leads to two rare genetic diseases, aplasia of lacrimal and salivary glands (ALSG: MIM #180929) and lacrimo-auriculo-dento-digital syndrome (LADD: MIM #149730)[13]. Both FGFR1b and FGFR2b are expressed in the epithelium during murine embryonic development and the ligands, FGF10 and FGF7, are expressed in the mesenchyme[14]. Paracrine signaling between mesenchymal FGF10 and epithelial FGFR2b is required and sufficient for SG initiation, while FGF7 is dispensable for gland initiation[15-17]. Reduced FGFR1b signaling, due to a hypomorphic mutation, leads to decreased branching morphogenesis and smaller glands[18]. Hypoplastic glands occur in $Fgf10^{+/-}$ and $Fgfr2b^{+/-}$ mice[19,20], while ligand-dependent-gain-of-function, due to an $Fgfr2b$ mutation ($Fgfr2^{+/Neo-S252W}$) leads to hyperplasia[21,22], suggesting sensitivity to FGF and FGFR protein expression levels. In addition, the SMGs of $FGFR2c^{+/\Delta}$ hemizygous mice were hypoplastic, highlighting a role for mesenchymal FGF signaling, although this is not the focus of this investigation[23]. Although SGs, like many other branched organs, are dependent on FGFR signaling for early development, the role of FGFRs in epithelial cell-specific lineages, progenitor function and differentiation of specific acinar cell types are not known.

SG development involves similar stages in both humans and mice, and the mouse SMG is a useful model to investigate both early development and cell differentiation[7,24,25]. Mouse SMG development begins at embryonic day (E) 11 as an epithelial invagination into a condensed mesenchyme, and at E12 a primary endbud enlarges and at E13 undergoes branching morphogenesis to give rise to all epithelial cells in the adult gland[26-29]. The differentiation of acinar and MECs begins ~E15, so that at birth, postnatal day 1 (P1), functional acinar, MEC and ducts result in salivary secretion[6,25]. Further postnatal maturation of acinar, MEC and ductal compartments occurs resulting in functional adult glands.

In this work, we investigate the role of FGFR signaling in specific SG cell types by leveraging existing single-cell RNA-sequencing (scRNAseq) datasets[30-33] and confirm that human and mouse SGs have similar expression patterns of both FGFRs and FGF ligands. We use mice carrying floxed FGFR alleles with epithelial Cre drivers to conditionally delete FGFRs in specific cell populations. We confirm the essential role of epithelial FGFR2 in the primary endbud for gland initiation and identify a requirement of FGFRs in adult duct progenitor function. We discover that FGFR2 signaling drives seromucous and serous acinar differentiation in the SMG and SLG, respectively, while FGFR1 is dispensable. Further investigation using ex vivo organ culture and loss and gain of function approaches identify that FGFR signaling occurs via MAPK signaling and the seromucous acinar transcriptional program is stimulated by FGF7, which is produced by adult MECs.

## Results

### Fgfr1 and Fgfr2 are enriched in postnatal basal duct, acinar and MECs

We propose that understanding the functional roles of FGFRs in specific adult cells in the SG will provide important mechanistic insight for developing therapeutic strategies. To this end, we first utilized the recently generated scRNAseq atlas[30] of murine SG development to map the general cell-specific expression of all FGFRs (Fig. 1a, b). The scRNAseq atlas contains several stages of embryonic (E12, E14 and E16) and postnatal development and we followed previous annotations of

this dataset[30]. There were no major differences in P30 and P300, and these stages were combined to increase cell numbers and referred to as adult in our analysis.

Both $Fgfr1$ and $Fgfr2$ were widely expressed in the embryonic epithelium with generally higher percentage expression at E12 compared to later stages in accordance with our previous reports[14, 34,35]. Specifically, $Fgfr1$ expression was detected in endbuds, (88% at E12, 78% at E14 to 40% at E16), Krt19+ ducts (40% at E12 and E14 to 20% at E16), and basal ducts (55% at E12 to 40% at E16, Fig. 1a). $Fgfr2$ was also detected in endbuds (50% at E12, 40% at E14 to 20% at E16), Krt19+ ducts (65% at E12, 20% at E14 to 10% at E16), and basal ducts (45% at E12 to 30% at E16) (Fig.1a). Expression in MECs was detected at E16, with 80% of cells expressing $Fgfr1$ and 40% expressing $Fgfr2$ (Fig. 1a). $Fgfr3$ was detected in only 5-7% of duct cells and $Fgfr4$ was barely detectable in E12 endbuds (~1%). Postnatally, $Fgfr1$ and $Fgfr2$ were both expressed in higher percentage of cells at P1, where the gland is undergoing postnatal growth and maturation, compared to adult glands. Specifically, in basal ducts the percentage of cells expressing $Fgfr$s decreased ($Fgfr1$: 55% at P1 to 17% in adult, $Fgfr2$: 32% at P1 to 4% in adult). A higher percentage of MECs expressed $Fgfr$s at P1 compared to adult ($Fgfr1$: 82% at P1 to 31% in adult, $Fgfr2$: 35% at P1 and 4% in adult). In acinar cells (P1 $Bpifa2+$, $Smgc+$, $mKi67+$ and adult serous or seromucous) the trends were similar with less percentage of cells expressing $Fgfr$s in adult compared to P1 ($Fgfr1$: ~26% at P1 to 1% in adult, $Fgfr2$: 23% at P1 to 5% in adult, Fig. 1b). Also, $Fgfr1$ was detected in 25% of $Gstt1+$ intercalated duct cells and $Fgfr2$ was expressed in 20% of P1 ionocytes (Fig. 1b), $Fgfr3$ was detected in 9% of basal duct cells and $Fgfr4$ was detected in <1% of basal ducts and acinar cells.

We also investigated FGFR expression using scRNAseq datasets from adult human SMG, MSG, and PG[31-33]. In general, human glands were similar to adult mouse SMG, with expression of FGFR1 and FGFR2 in basal duct (FGFR1: 5-40%, FGFR2: 30-50%) and MECs (FGFR1: 50-70%, FGFR2: 30-40%). Acinar cells also showed expression of FGFR2 (2-60%) and some expression of FGFR1 (1-5%). Similar to mouse SMG, basal cells additionally expressed FGFR3 (11-30% of cells in clusters) and FGFR4 was detected in 2% or less of basal duct cells (Supplementary Fig. 1a).

We focused our analysis on $Fgfr1$ and $Fgfr2$ due to their higher expression in both embryonic and postnatal development, compared to other $Fgfr$ genes. Since embryonic expression patterns have previously been reported[14,35], we focused on confirming cell-specific expression patterns in postnatal glands. RNAscope in situ hybridization was used to highlight the overlap of FGFR expression with known cell types in postnatal glands, although the actual transcript number of each FGFR in individual cells of a specific cell type may be different. In situ hybridization in postnatal glands, confirmed the enrichment in $Krt5+$ cells, which include basal ducts and MECs (Fig. 1c, Supplementary Fig. 1b). There was also widespread $Fgfr1$ and $Fgfr2$ coexpression with the acinar marker $Bhlha15+$ (MIST1, Fig. 1d). Notably, at P1, acinar cells were enriched for either $Fgfr1$ or $Fgfr2$, although not exclusive, while adult acinar cells were overall enriched for both receptors (Fig. 1d).

Taken together, $Fgfr1$ and $Fgfr2$ are the most widely expressed epithelial FGFRs in SGs, enriched in MECs, basal duct and acinar cells, all populations that have progenitor potential during regeneration and self-renewal during adult homeostasis. Based on these findings, we predicted that both FGFR1 and FGFR2 signaling are important for SG epithelial development and critical in specific lineages and cell populations at later developmental stages. We sought to test this hypothesis using bioinformatic data to direct studies with genetic mouse models and explant culture.

### Distinct roles of Fgfr1 and Fgfr2 in SG, limb and craniofacial development

Based on previous global FGFR knockout studies showing differential roles for FGFR1 and FGFR2 during development, we predicted that

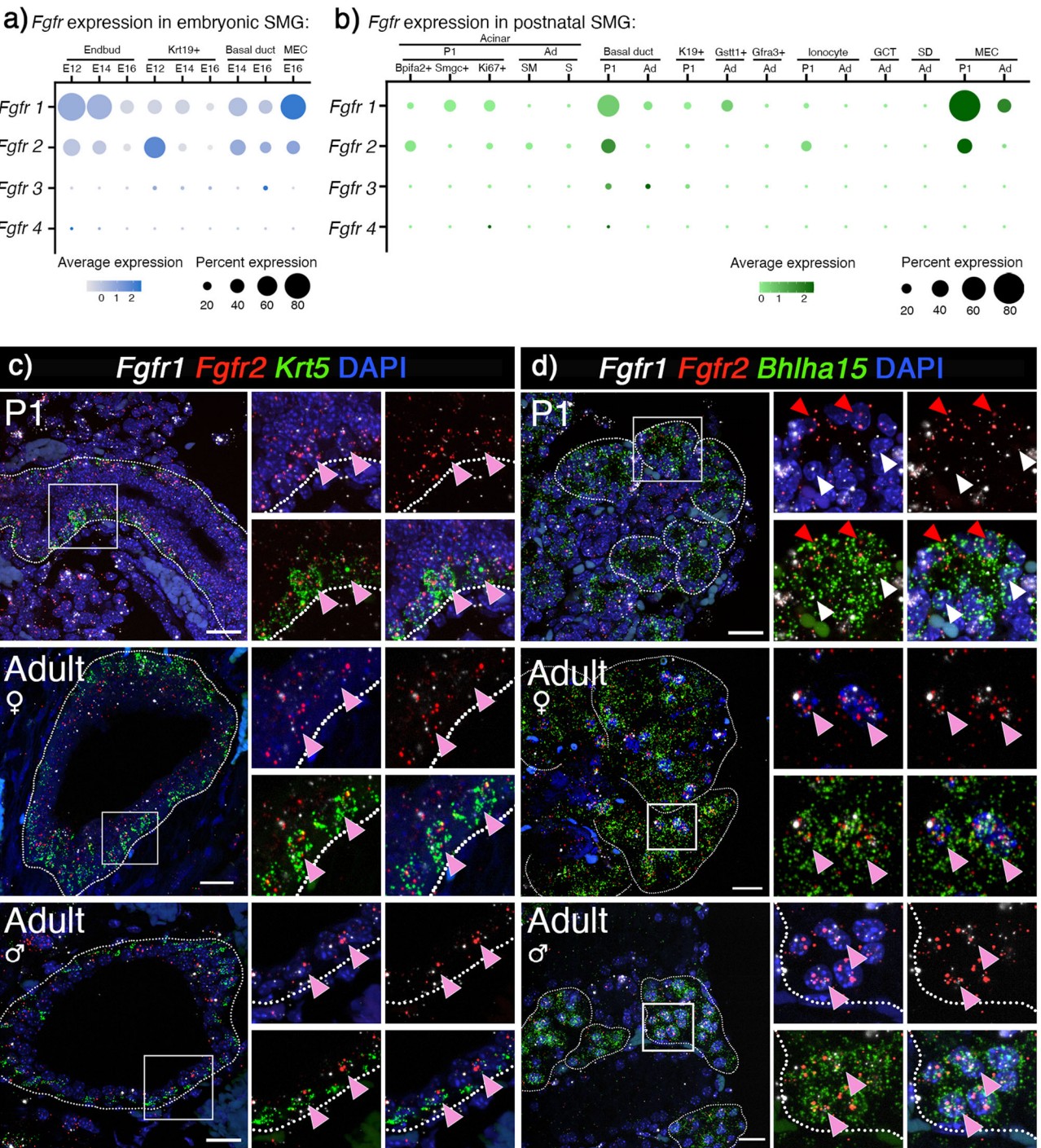

**Fig. 1 | Epithelial *Fgfr1* and *Fgfr2* are broadly expressed in mouse SMG and enriched in basal duct, acinar and MECs during postnatal development.**
**a** Dotplot of scRNAseq data show that *Fgfr1* and *Fgfr2* were enriched in epithelial cell populations at E12. At E14 and E16, expression of both receptors decreased in Krt19+ duct and endbud, while it is maintained in basal duct and MECs. Fgfr: Fibroblast growth factor, SMG: Submandibular gland, E: Embryonic day. **b** *Fgfr1* and *Fgfr2* were enriched in acinar cells, basal duct, and MECs in postnatal glands. P1: postnatal day 1, Ad: Adult. Subclusters of P1 acinar cells enriched for *Bpifa2*, *Smgc* or m*Ki67* and adult acinar cells that are seromucous (SM) and serous (S) are shown.

P: Postnatal day. **c** Representative images showing in situ hybridization with *Fgfr1* (white), *Fgfr2* (red), and *Krt5* (green) showed enrichment in basal duct cells in SMGs from P1 and adult female and male (pink arrowhead). Scale bar: 20 μm.
**d** Representative images showing in situ hybridization with *Fgfr1* (white), *Fgfr2* (red), and *Bhlha15* (green) showed enrichment in acinar cells at both P1 and adult female and male. P1 acinar cells were enriched for *Fgfr1* (red arrowhead) or *Fgfr2* (white arrowhead), while adult acinar cells were equally enriched for both receptors (pink arrowhead). Scale bar: 20 μm.

deleting *Fgfr1* in the entire E12 epithelium would not affect gland initiation, while *Fgfr2* would be required. Endbud and duct cells in the E12 epithelium can be identified bioinformatically by enrichment of *Sox10* and *Krt5*, respectively (Fig. 2a, b)[26,36]. At this stage, both populations expressed *Fgfr1* and *Fgfr2* along with the ectodermal

transcription factor AP-2 (*Tfap2a*) and the epithelial marker *Krt14* (Fig. 2b). Based on this, we used the ectoderm-specific *Tfap2aCre* (Crect) and the epithelial specific *Krt14Cre* mouse strains, crossed with mice carrying floxed alleles for *Fgfr1* and *Fgfr2* in addition to the cell membrane-targeted, two-color fluorescent Cre-reporter mTmG mouse

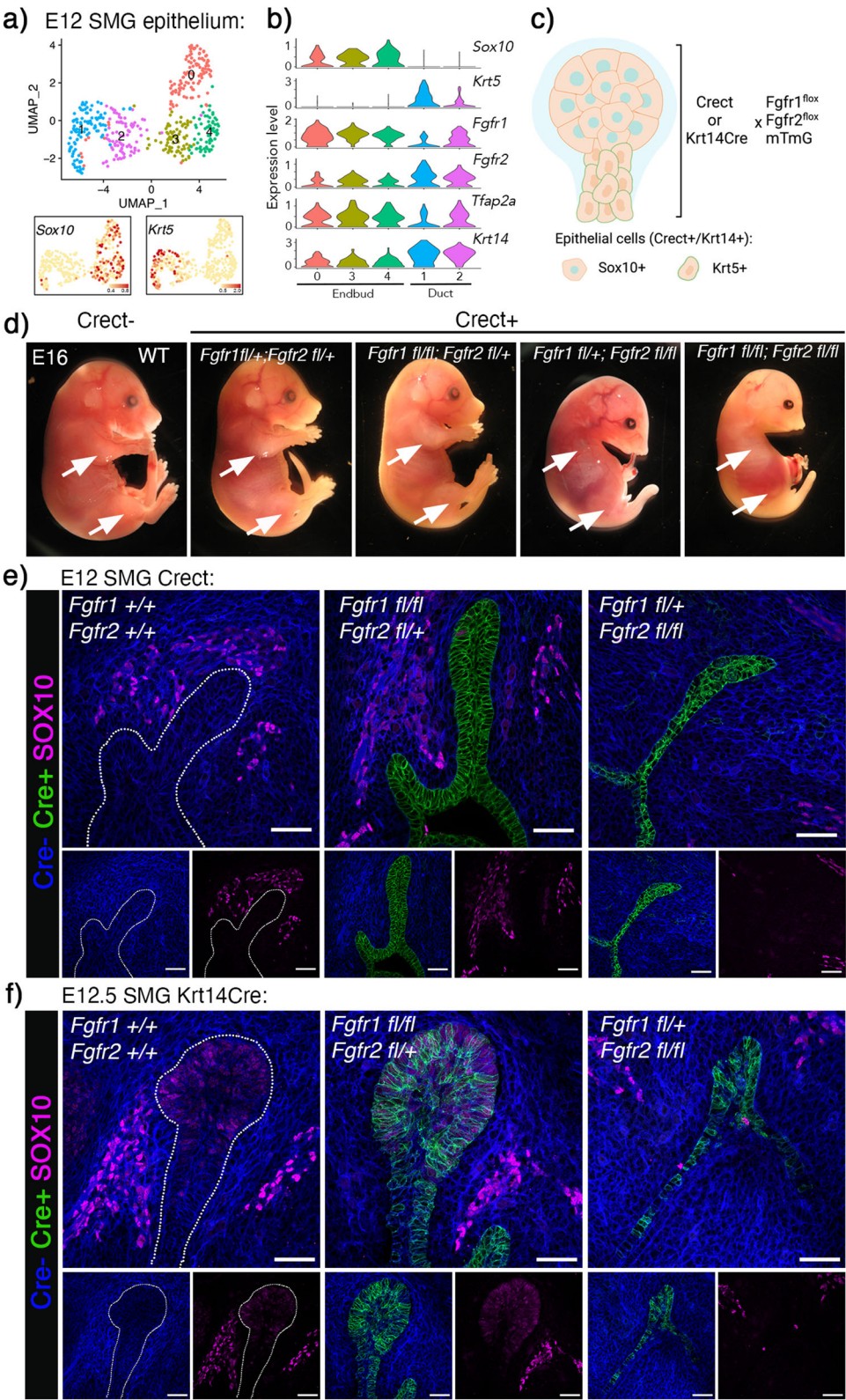

**Fig. 2 | *Fgfr1 and Fgfr2* are required for SMG and craniofacial development.**
**a** UMAP showing E12 SMG epithelium grouped into endbud (clusters 0, 3 and 4) and duct (clusters 1 and 2) based on *Sox10* and *Krt5* expression, respectively. **b** Violin plots showing enrichment of *Fgfr1, Fgfr2, Tfap2a (Crect), and Krt14* in Sox10+ endbud and Krt5+ duct. c) Strategy for *Fgfr1* and *Fgfr2* deletion in SG epithelium. Created using Biorender.com. d) Gross images of *Crect; Fgfr1ᶠˡ;Fgfr2ᶠˡ* embryos at E16. *Crect + ;Fgfr1ᶠˡ + ;Fgfr2ᶠˡ/ᶠˡ* embryos failed to develop limbs (arrows) and had cleft palate and maxillar and mandibular hypoplasia. This was exacerbated

upon the loss of an additional *Fgfr1* allele. Embryos that are heterozygous for *Fgfr2* were grossly normal and indistinguishable from *Crect-* littermate controls. **e** Deletion of *Fgfr1* and one copy of *Fgfr2* was comparable to WT glands, while *Fgfr2* deletion in Crect+ epithelial cells impede SMG and SLG development. Representative images show Cre + : green, Cre-: blue, SOX10: magenta. Scale bars: 50 μm. **f** Deletion of *Fgfr1* in Krt14+ cells did not affect endbud formation, while *Fgfr2* deletion led to loss of Sox10+ endbud cells. Representative images show Cre + : green, Cre-: blue, SOX10: magenta. Scale bars: 50 μm.

strain, resulting in *Fgfr* deletion in vivo (Fig. 2c). This generates embryos where Cre+ cells have epithelial *Fgfr* deletions and express cell membrane-localized GFP. Thus, for further analysis, GFP was used as a pseudomarker for epithelial *Fgfr* manipulation. We predicted both *Crect* and *Krt14Cre* would result in similar GFP activation within the oral cavity and confirm the ectodermal origin of SG epithelium, although differential temporal expression patterns of the two Cre drivers would result in various defects in other tissues.

While *Crect + ;Fgfr1*$^{fl/+}$*;Fgfr2*$^{fl/+}$ embryos were indistinguishable from *Crect-* controls, *Crect + ;Fgfr1*$^{fl/+}$*;Fgfr2*$^{fl/fl}$ embryos strongly phenocopied other *Fgfr2* knockout transgenic mice and were non-viable, likely due to severe craniofacial defects upon *Fgfr2* deletion. The embryo weights were comparable to control, but failed to develop limbs, had fused caudal vertebrae, cleft palate and maxillary and mandibular hypoplasia (Fig. 2d, Supplementary Fig. 2a, b). Interestingly, the phenotype was exacerbated in the *Crect + ;Fgfr1*$^{fl/fl}$*;Fgfr2*$^{fl/fl}$ mice, suggesting additional roles for *Fgfr1* in ectodermal tissues, especially for mandibular and craniofacial development (Fig. 2d). Deleting FGFRs in the Krt14+ lineage resulted in a milder phenotype; limbs developed in *Krt14Cre + ;Fgfr1*$^{fl/+}$*;Fgfr2*$^{fl/fl}$, but defects in digitization of the hind limbs and eyelid formation were observed in E16 embryos (Supplementary Fig. 2c). Interestingly, there were no obvious craniofacial developmental defects with loss of *Fgfr2* in the Krt14+ lineage. The *Krt14Cre + ;Fgfr1*$^{fl/fl}$*;Fgfr2*$^{fl/+}$ also appeared similar to *Krt14Cre-* controls up to E16 (Supplementary Fig. 2c).

SGs from the E12 *Crect + ;Fgfr1*$^{fl/fl}$*;Fgfr2*$^{fl/+}$ were comparable to control and formed endbuds with a stratified, invaginating epithelium in the condensing mesenchyme, confirming the ectodermal origin of salivary epithelium (Fig. 2e). In embryos with *Fgfr2* deletion, initiation of both the SMG and SLG occurred, appearing as an infolding of the epithelium. The fold in the oral epithelium was similar to an early E11 SMG but failed to form a stratified endbud (Fig. 2e). Similarly, *Krt14Cre + ;Fgfr1*$^{fl/fl}$*;Fgfr2*$^{fl/+}$ glands were comparable to control while *Krt14Cre + ;Fgfr1*$^{fl/+}$*;Fgfr2*$^{fl/fl}$ glands had a thickening of the oral epithelium but had not stratified to form an endbud, further shown by loss of SOX10+ endbud cells (Fig. 2f). Mesenchymal SOX10+ expression in neural crest cells, precursors to the parasympathetic ganglion, was not affected (Fig. 2f).

Together, these data confirm the central role of *Fgfr2* in gland initiation and demonstrate that *Fgfr1* is dispensable for endbud formation. Further, to investigate cell specific roles in lineage specification, FGFRs were deleted in duct and acinar cells directly using additional cre models.

**Postnatal duct progenitor function is *Fgfr1* and *Fgfr2* dependent**
Next, we aimed to determine whether the expression of FGFRs in basal ducts is required for their development. At E12, *Sox10* expressing endbud cells give rise to the gland parenchyma, while KRT5 has a more restricted expression pattern in the oral epithelium and SG duct as previously reported[26, 36]. Thus, we predicted that loss of *Fgfr1* and *Fgfr2* in the E12 Krt5+ ducts might be compensated for by Sox10+ lineage.

Bioinformatic analysis of duct cells from E14, E16, and P1 showed a clear enrichment of *Fgfr1* and *Fgfr2* in basal duct, while luminal ducts were enriched for *Krt19*, *Cldn3*, *Cldn4*, and *Foxi1* (Supplementary Fig. 2d). Accordingly, epithelial *Fgfr1* and *Fgfr2* were specifically deleted in basal duct lineage by crossing the *Fgfr1;Fgfr2* floxed mice with a constitutive *Krt5Cre* strain (Fig. 3a). In this model, *Fgfr1* and *Fgfr2* are deleted in E12 Krt5+ duct lineage, while being expressed elsewhere in the gland (Fig. 3a, Supplementary Fig. 2d). Previous work has shown that this cross leads to viable mice; however, they develop a progressive skin phenotype[37]. In addition, the SGs were significantly smaller (~40%) in adult *Krt5Cre + ;Fgfr1*$^{fl/fl}$*;Fgfr2*$^{fl/fl}$ mice compared to wildtype (WT, Krt5Cre-, Supplementary Fig. 2e). The *Krt5Cre + ;Fgfr*$^{fl/fl}$*;Fgfr2*$^{fl/fl}$ SGs had a distinct duct phenotype with 60% reduction duct/total gland area and

loss of granular convoluted tubules (GCT), which are sexually dimorphic, specialized ducts producing NGF and EGF in mice (Fig. 3b, c). These results points to a specific role for FGFR signaling in basal progenitors either during duct development or postnatal GCT differentiation.

To further dissect the temporal role of FGFRs in duct lineages, we used the inducible *Krt5rtTA;tetCre* mice crossed with *Fgfr1*$^{fl/fl}$*;Fgfr2*$^{fl/fl}$ and mTmG strains to investigate whether the observed phenotype in the non-inducible model is due to pre or postnatal events. These mice carry a tetracycline transactivator gene that induces Cre expression in *Krt5+* basal cells. We analyzed the effect of FGFR deletion in the Krt5+ lineage during embryonic duct development by feeding doxycycline to pregnant females and harvested glands from pups at P1. Additionally, we studied postnatal duct lineage by performing lineage tracing experiments in adult mice.

Deletion of *Fgfr1* and *Fgfr2* from the Krt5+ lineage during embryonic development resulted in no significant change in P1 gland weight normalized to body weight (Supplementary Fig. 2f). Gross histology was normal and gene expression analysis showed a reduced trend in ductal markers although not significant in *Krt5rtTA;tetCre + ;Fgfr1*$^{fl/fl}$*;Fgfr2*$^{fl/fl}$ glands compared to WT (Cre-) (Supplementary Fig. 2g, h). This suggests that FGFR signaling in embryonic Krt5+ basal cells is either not involved in lineage contribution and differentiation of duct populations, or that loss of FGFR signaling in Krt5+ cells can be compensated for by other pathways. It also suggests that the duct phenotype observed in the non-inducible model was likely due to postnatal differentiation events.

To address the potential role of FGFR1 and FGFR2 in basal duct lineage contribution to GCTs, adult *Krt5rtTA;tetCre + ;Fgfr1*$^{fl/fl}$*;Fgfr2*$^{fl/fl}$*; mTmG* mice (male and female, 6-8 weeks) were fed doxycycline for 4 days (pulse, day 0) resulting in FGFR deletion and onset of GFP reporter expression (Fig. 3d). After FGFR deletion, we predicted that lineage tracing similar to control would indicate FGFR independence of progenitor function, while decreased lineage contribution would indicate the opposite. After a 90-day chase, lineage tracing from Krt5+ duct cells was clearly evident in control glands (*Krt5rtTA;tetCre + ;Fgfr1*$^{fl/+}$*;Fgfr2*$^{fl/+}$*;mTmG*), while this pattern was not seen in glands from *Krt5rtTA;tetCre + ;Fgfr1*$^{fl/fl}$*;Fgfr2*$^{fl/fl}$*;mTmG* mice (Fig. 3e). Additionally, non GFP+ duct cells exhibited abnormal morphology after a 90-day chase in *Krt5rtTA;tetCre + ;Fgfr1*$^{fl/fl}$*;Fgfr2*$^{fl/fl}$*;mTmG* glands, suggesting a disruption of duct integrity or homeostasis, or alternatively loss of a paracrine factor secreted by FGFR expressing duct cells, although further work is needed to confirm this. Quantification of GFP expression showed similar baseline levels in both control and with *Fgfr1;Fgfr2* deletion, although expression in males was higher than females (Fig. 3f and Supplementary Fig. 2i) likely due to the prominence of GCT ducts in male murine SGs and potentially suggesting FGFRs may have a role in GCT homeostasis. While control glands had significant increase in GFP expression after 90 days chase, no significant changes were seen in *Krt5rtTA;tetCre + ;Fgfr1*$^{fl/fl}$*;Fgfr2*$^{fl/fl}$*;mTmG* glands (Fig. 3f). These results show that FGFR1 and FGFR2 signaling is required for basal progenitor contribution to duct homeostasis in adult male and female SMGs (Fig. 3g).

***Fgfr1* and *Fgfr2* enrichment in developing acinar subpopulations**
Next, we investigated whether FGFRs are required for acinar differentiation because our initial analysis showed *Fgfr1* and *Fgfr2* expression in both embryonic endbuds and postnatal acinar cells (Fig. 1a, d). Acinar specification starts around E15 with onset of the canonical acinar markers *Aqp5*, *Bhlha15* and *Cldn10*[30,38,39], and adult acinar cells are defined as serous, seromucous or mucous based on their secretory products. In P1 SMGs, two subpopulations of acinar cells are defined by *Bpifa2* and *Smgc*, and their transcriptional profiles are overlapping with mature seromucous acinar cells and adult *Gstt1+* intercalated ducts, respectively[30].

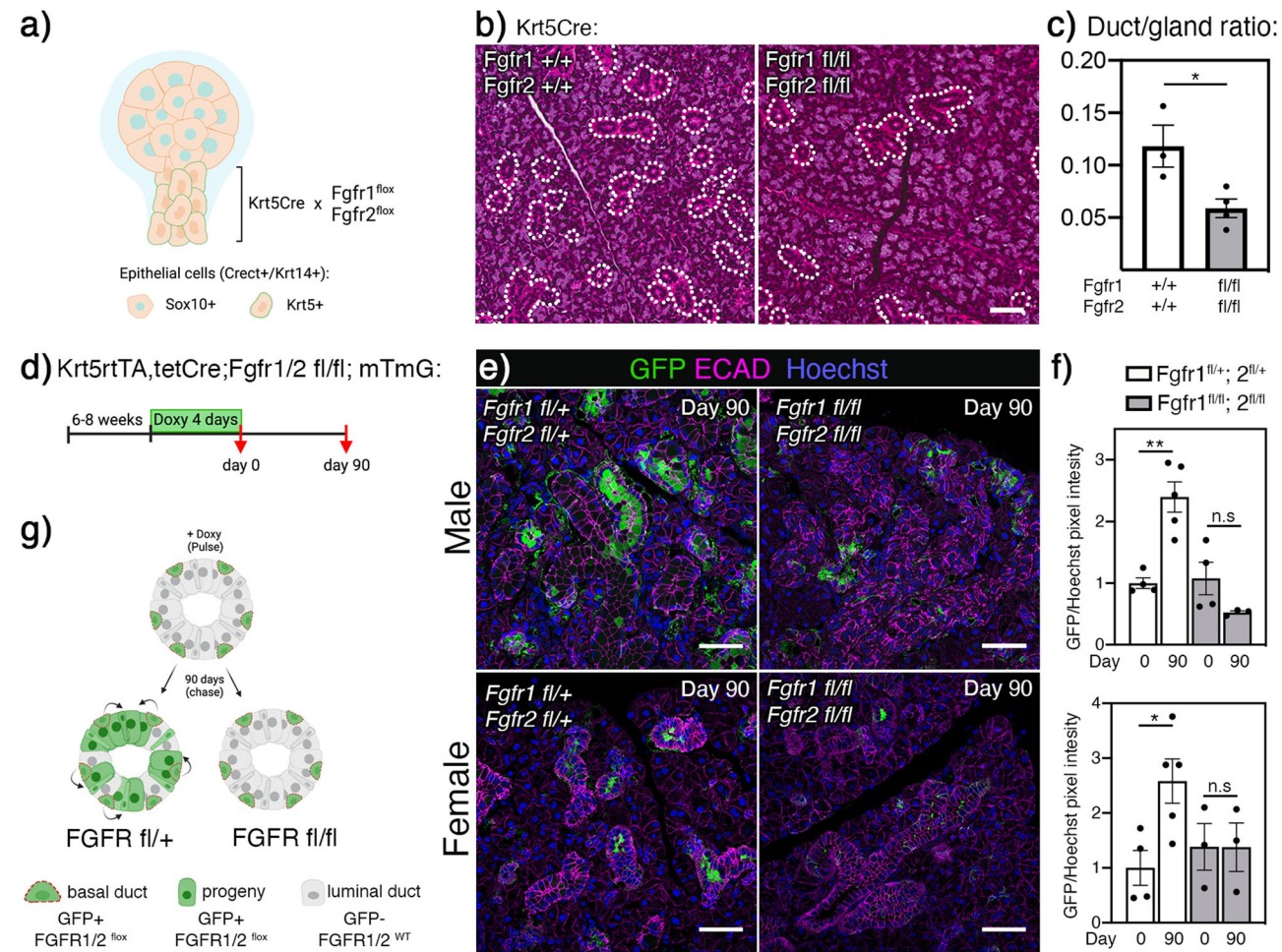

**Fig. 3 | *Fgfr1* and *Fgfr2* are required for postnatal duct development and lineage contribution. a** *Krt5*Cre mice were crossed with *Fgfr1^fl^;Fgfr2^fl^* strains to delete FGFRs in *Krt5* lineage during development. Created using Biorender.com. **b** Representative images of H&E staining from WT and *Krt5*Cre + ;*Fgfr1^fl/fl^;Fgfr2^fl/fl^* SMGs. Dotted lines indicate duct area. Scale bars: 50 μm. **c** Quantification of duct/gland ratio showed reduction in duct area after *Fgfr1* and *Fgfr2* deletion in Krt5+ lineage in adult SMGs (WT *n* = 3 and *Krt5*Cre + ;*Fgfr1^fl/fl^;Fgfr2^fl/fl^ n* = 4). Data are presented as mean values±SEM. Unpaired two-tailed t-test was used to calculate significance (*p* = 0.0298). Source data are provided as a Source Data file. **d** Adult (6–8 weeks old) *Krt5rtTA;tetCre;Fgfr1/2^fl^; mTmG* mice were fed doxycycline for 4 days (pulse, day 0) and glands were analyzed after a 90-day chase. **e** SMGs sections were stained for E-cadherin (ECAD, magenta) and GFP (green) and Hoechst (blue). Day 90 showed less GFP expression in samples where *Fgfr1* and *Fgfr2* were deleted compared to control. Scale bars: 50 μm. **f** Quantification of GFP expression in IHC sections at day 0 and day 90 showed reduced contribution from Krt5+ duct cells after *Fgfr1* and *Fgfr2* deletion in both male and female glands. Unpaired two-tailed t-test was used to calculate significance between day 0 and day 90 for each genotype (*p* = 0.0215, **p* = 0.0018, ns = not significant). Males: *Fgfr1/2^fl/+^ n* = 4 (day 0) and *n* = 5 (day 90), *Fgfr1/2^fl/fl^ n* = 4 (day 0) and *n* = 3 (day 90). Females: *Fgfr1/2^fl/+^ n* = 4 (day 0 and day 90), *Fgfr1/2^fl/fl^ n* = 3 (day 0 and day 90). Data are presented as mean values±SEM. Source data are provided as a Source Data file. g) Increased GFP expression after 90 days chase was evident in control group, while FGFR deletion led to no significant lineage contribution suggesting progenitor function in adult basal cells is FGFR dependent. Created using Biorender.com.

To further analyze FGFR expression at previously defined stages of acinar differentiation, we bioinformatically isolated and re-clustered E16 and P1 acinar cells (Fig. 4a and Supplementary Fig. 3a). This allowed detection of cell specific genes during the establishment and differentiation of acinar subpopulations. Clusters 0 and 3 correspond to *Smgc*+ acinar cells and were enriched for *Fgfr1*, while cluster 1 corresponded to *Bpifa2*+ acinar cells and were enriched for *Fgfr2* (Fig. 4b). *Fgfr1* + :*Smgc*+ cells were also enriched for *Gstt1, Ramp1, Cdkn1c, Tesc* and *Lman1* while *Fgfr2* + :*Bpifa2*+ cells were enriched for *Mucl2, Lpo, Dcpp1, Car6, Prol1* and *Elf5* (Fig. 4c). In general, all markers had higher expression level at P1 compared to E16. *Smgc, Bpifa2* and *Lpo* were enriched in their respective populations at both E16 and P1; however, some markers were stage-specific, such as *Tesc* at E16 and *Gstt1* at P1 in *Fgfr1*+ cells and *Mucl2, Car6* and *Prol1* in *Fgfr2*+ cells at P1 (Supplementary Fig. 3b).

Based on average expression, cluster 2 (Fig. 4c) appeared to be double positive for *Fgfr1* and *Fgfr2*; however, this cluster contained either *Smgc*+ or *Bpifa2*+ cells, and was defined by proliferative markers such as *Mki67, Bub1b, Aurka* and *Top2a* (Supplementary Fig. 3c). Pathway analysis indicated active proliferation as the major functional state of these cells (Supplementary Fig. 3c). Cluster 4 expressed relatively low levels of the canonical acinar markers and had additional genes from multiple cell lineages that do not align with acinar cell identity including, but not limited to, *Krt14, Krt5, Trp63, Acta2, Col1a1* and *Col3a1*. Although doublets were previously removed from the dataset, it is not clear whether this small cluster are remaining doublets or cells in a transitional cell state[4]. Due to this ambiguity, and to focus specifically on acinar cells, this cluster was excluded from further analysis.

To verify the timeline for acinar specification (onset of canonical acinar markers) together with differentiation (specific acinar subpopulations) using previously established markers and genes we had identified in specific *Fgfr* enriched populations, we performed qPCR of embryonic SMGs at E14, E15 and E16, detecting acinar specification by

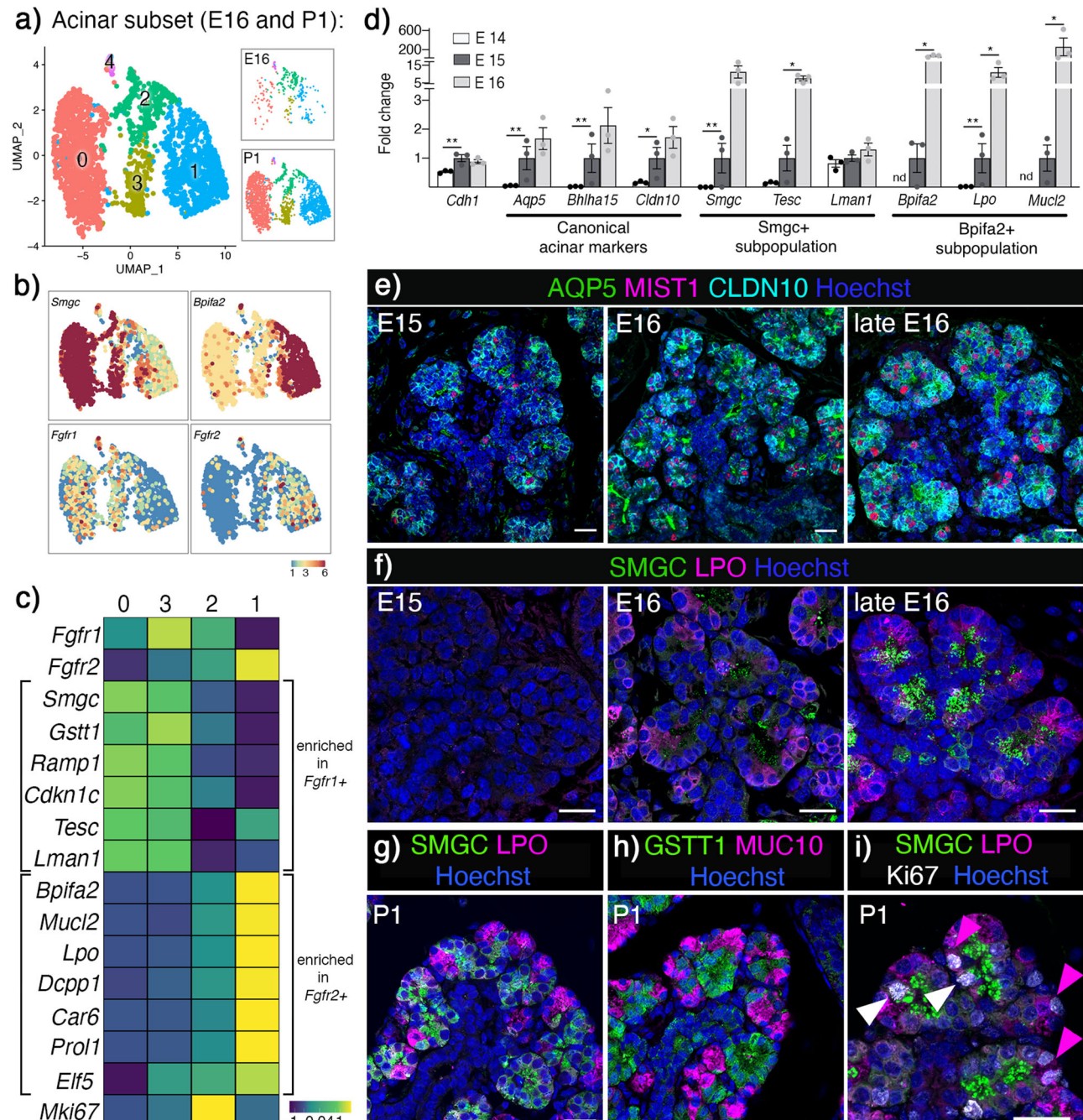

**Fig. 4 | *Fgfr1* and *Fgfr2* are differentially enriched in specific acinar subpopulations during gland development. a** UMAP showing E16 and P1 acinar cells. This dataset was used to analyze *Fgfr* expression and transcriptional profiles of the acinar subpopulations. **b** UMAPs showing expression of *Smgc*, *Bpifa2*, *Fgfr1*, and *Fgfr2* in E16 and P1 acinar cells. **c** Heatmap showing average expression of genes enriched in *Fgfr1+* and *Fgfr2+* acinar populations. **d** Onset of acinar markers and subpopulations in vivo occurred around E15 ($n = 3$, normalized to E15 and house-keeping gene *Rsp29*, nd=not detected). One-way ANOVA with Dunnett's test for multiple comparisons (*Cdh1* \*\*$p = 0.006$, *Aqp5* \*\*$p = 0.0037$, *Bhlha15* \*\*$p = 0.0058$, *Cldn10* \*$p = 0.02$, *Smgc* \*\*$p = 0.0072$, *Tesc* \*$p = 0.0156$, *Bpifa2* \*$p = 0.0467$, *Lpo* \*\*$p = 0.0099$, \*$p = 0.0221$, *Mucl2* \*$p = 0.0187$). Data are presented as mean values ±

SEM. Source data are provided as a Source Data file. **e** Representative images showing acinar markers AQP5 (Aquaporin5, green), MIST1(magenta) and CLDN10 (Claudin10, cyan) and Hoechst (nuclei, blue) could be detected from E15. Scale bars: 50 μm. **f** Representative images showing protein expression of SMGC (Sub-mandibular gland protein C, green), LPO (Lactoperoxidase, magenta) and Hoechst (nuclei, blue) was detected by E16. Scale bars: 20 μm. **g, h** Representative image showing the two populations either by SMGC (green) and LPO (magenta) or GSTT1 (Gluthathione S-transferase theta 1, green) and MUC10 (Mucin10, magenta) in P1 SMG. Nuclei are stained with Hoechst (blue). Scale bars: 20 μm. **i** Proliferating Ki67+ cells (white) were detected in both SMGC (green) and LPO (magenta) populations in P1 SMG. Representative image, Hoechst stained nuclei (blue), scale bar: 20 μm.

E14 and differentiation of both subpopulations by E15 (Fig. 4d). Immunostaining at E15 showed progressively increasing organization of luminal AQP5, basolateral CLAUDIN10 (CLDN10) and nuclear MIST1 (Fig. 4e). Markers for the two subpopulations were detected from E16, and by late E16 both populations were clearly distinguished with

immunostaining by SMGC and LPO proteins (Fig. 4f). In P1 glands, the two acinar populations were visualized through immunostaining with either SMGC and LPO or GSTT1 and MUC10 (Fig. 4g, h). Co-staining of SMGC and GSTT1 or LPO and MUC10 further confirmed overlapping expression of these markers (Supplementary Fig. 3d, e). Additionally,

proliferating acinar cells were found within both subpopulations (SMGC and LPO), further suggesting that cluster 2 is mainly proliferating cells made up by a mix of the two distinct populations rather than a unique population (Fig. 4i).

Taken together, *Fgfr1* and *Fgfr2* were enriched within specific acinar subpopulations that could be identified by specific markers and our analysis highlights E16 and P1 as appropriate stages to investigate the establishment of both subpopulations. We hypothesized that FGFR signaling would differentially affect expression of defining markers and either the development or maturation of these acinar subpopulations.

### *Fgfr2* is required for differentiation of seromucous acinar cells

To test the direct effect of FGFR signaling on acinar differentiation, we used the Aqp5Cre-IRES-dsRed mouse strain (ACIDCre) in combination with *Fgfr1*$^{fl/fl}$;*Fgfr2*$^{fl/fl}$;*tdTomato* mice, targeting epithelial *Fgfr1* and *Fgfr2* in acinar cells. AQP5 is expressed in several tissues including the lung[40] and pups did not survive more than three days after birth with *Fgfr2* deletion in AQP5+ cells. *ACIDCre +;Fgfr2*$^{fl/fl}$ animals were only found among P1 pups and no adult mice with this genotype could be generated (Supplementary Fig. 4a). Due to this phenotype, we analyzed acinar development between P1 and P3 in this model.

In ACIDCre+ SMGs, gross histology of *Fgfr1*$^{fl/+}$;*Fgfr2*$^{fl/+}$ and *Fgfr1*$^{fl/fl}$;*Fgfr2*$^{fl/+}$ were comparable to control, while both *Fgfr1*$^{fl/+}$;*Fgfr2*$^{fl/fl}$ and *Fgfr1*$^{fl/fl}$;*Fgfr2*$^{fl/fl}$ had pronounced acinar hypoplasia (Fig. 5a). This was reflected in gland size, where both *Fgfr1*$^{fl/+}$;*Fgfr2*$^{fl/fl}$ and *Fgfr1*$^{fl/fl}$;*Fgfr2*$^{fl/fl}$ were ~40% smaller compared to WT (Fig. 5b). Still, gene expression of *Aqp5, Bhlha15, and Cldn10* were detected in all genotypes (Fig. 5c). Accordingly, AQP5, MIST1 and CLDN10 expressing cells were found in all genotypes, although acinar organization was severely disrupted and MIST1+ cells per area was significantly reduced after *Fgfr2* deletion (Fig. 5d, h). After *Fgfr2* deletion, there was no change in the defining genes for *Fgfr1*+ cells (*Smgc, Gstt1,* or *Ramp1*), while expression of *Bpifa2, Prol1, Mucl2,* and *Elf5* was significantly reduced with a similar trend for *Lpo* (Fig. 5c). Interestingly, *Bpifa2, Prol1, and Mucl2* were also decreased in *Fgfr1*$^{fl/+}$;*Fgfr2*$^{fl/+}$ and *Fgfr1*$^{fl/+}$;*Fgfr2*$^{fl/+}$. This was due to heterozygous *Fgfr2* rather than *Fgfr1* deletion as *Fgfr1*$^{+/+}$;*Fgfr2*$^{fl/+}$, gave similar results (comparison shown in Supplementary Fig. 4b). Protein detection of the two populations showed loss of LPO and MUC10 after *Fgfr2* deletion, while SMGC and GSTT1 were detected in all genotypes (Fig. 5e, f). Furthermore, all acinar cells (MIST1 + ) expressed SMGC after *Fgfr2* deletion, highlighting the complete loss of seromucous differentiation (Fig. 5g). When focusing on seromucous acinar proteins, quantification showed a significant decrease in LPO, while gene expression was still detected suggesting post-transcriptional regulation (Fig. 5i). Similarly, MUC10 (*Prol1*) staining was decreased after *Fgfr2* deletion, consistent with its gene expression (Fig. 5i).

SLG acini consist of MUC19+ mucous cells and LPO+ serous cells. FGFR2 signaling was important in the SLG as evidenced by the loss of serous cells after *Fgfr2* deletion (Supplementary Fig. 4c–e). *Fgfr2* deletion did not change, *Aqp5* and *Bhlha15*, while serous markers *Lpo, Bpifa2* and *Dcpp1* were reduced (Supplementary Fig. 4d). Interestingly, *Sox2*, a potency marker in SLG acinar cells was increased in expression, suggesting there may be increased SOX2+ progenitors when serous acinar differentiation is reduced due to lack of FGFR2 signaling (Supplementary Fig. 4d). Immunostaining supported this finding and there was a clear decrease of LPO staining in *Fgfr1*$^{fl/+}$;*Fgfr2*$^{fl/fl}$ glands while MUC19 staining in mucous cells was evident in all genotypes (Supplementary Fig. 4e). Taken together, these data show that *Fgfr1* is dispensable while *Fgfr2* is required for differentiation and expression of secretory markers in seromucous and serous cells in the SMG and SLG, respectively. It also highlights that mucous acinar cell differentiation is independent of FGFR signaling.

### *Fgfr2* via MAPK is required for seromucous acinar differentiation

We utilized WT mice (ICR) and an organ culture system to further manipulate downstream signaling required for FGFR-dependent seromucous acinar differentiation. E15 SMGs were cultured for 24 and 48 h to establish baseline gene expression during culture conditions (Supplementary Fig. 5a). This showed consistent expression of *Fgfr1b* and *Fgfr2b* using isoform specific PCR primers, *Fgf10* and *Fgf7*, and the downstream effectors *Etv4*, and *Etv5* (Supplementary Fig. 5b). Acinar differentiation ex vivo reiterated in vivo gene expression profiles of *Aqp5, Bhlha15* and *Cldn10* as well as markers for the two subpopulations including *Smgc, Tesc, Lman1, Bpifa2, Lpo,* and *Mucl2* (Supplementary Fig. 5c). Acinar cells were evident by staining of AQP5, MIST1 and CLDN10 and the two populations could be visualized by SMGC and LPO after 24hrs in culture, comparable to in vivo localization patterns (Supplementary Fig. 5d, e). For loss-of-function experiments, E15 SMGs were cultured for 24 h with chemical signaling inhibitors targeting canonical downstream pathways (Fig. 6a). Based on inhibitor concentrations used in SMG organ cultures in previous reports, we treated glands with a recombinant mouse FGFR2b-Fc protein (rFGFR2b, 20 µg/ml) to compete endogenous FGFs, a pan-FGFR inhibitor (SU5402, 5 µM), a MAPK inhibitor (UO126, 20 µM), a PI3K inhibitor (Ly249002, 25 µM), a PLCγ inhibitor (U73122, 25 µM) or vehicle control (BSA or DMSO) for 24 h. We initially analyzed apoptosis (Cleaved-caspase3) and proliferation (Ki67) by immunostaining after inhibitor treatment. Both rFGFR2b and pan-FGFR inhibitor treatment upregulated apoptosis with a concomitant reduction in proliferation compared to control (Fig. 6b, c). Treatments inhibiting either MAPK or PI3K inhibitors did not affect proliferation or apoptosis, while PLCγ inhibition significantly increased proliferation (Fig. 6b, c). These data show that changes in gene expression are not directly due to changes in cell survival and proliferation after MAPK or PI3K inhibitor treatment.

Expression of *Cdh1, Fgfr1b, Fgfr2b, Fgf10, Fgf7, Smgc, Tesc* and *Lman1* was not affected by any of the inhibitors tested, except rFGFR2b which increased *Fgf10* (Supplementary Fig. 5f and Fig. 6d). Expression of the FGFR downstream effectors, *Etv4* and *Etv5*, was reduced after rFGFR2b, pan-FGFR, and MAPK inhibitor treatment, confirming inhibition of the FGFR signaling pathway (Supplementary Fig. 5f). There was a trend of reduced expression of *Bhlha15, Cldn10,* and *Aqp5* expression after rFGFR2b, pan-FGFR and MAPK inhibitor treatments. Although, *Aqp5* expression was significantly reduced after MAPK inhibition, indicating a specific MAPK signaling dependence for *Aqp5*, and also *Cldn10* was significantly decreased after FGFR inhibition (Fig. 6d). Both *Lpo* and *Bpifa2* expression was decreased after rFGFR2b, pan-FGFR, MAPK and PI3K inhibitor treatment, whereas PLCγ inhibitor treatment did not affect *Fgfr2*-dependent gene expression (Fig. 6d). In addition, SGs treated with rFGFR2b showed decreased phosphorylated ERK, indicating the gene expression changes detected after rFGFR2b treatment were due to decreased MAPK pathway activity (Supplementary Fig. 5g). Gross histology of all treatment groups was comparable to control after 24 h treatment (Supplementary Fig. 5h) and luminal AQP5, nuclear MIST1 and lateral CLDN10 were detected in all groups, indicating that acinar specification was not affected (Fig. 6e). Staining for SMGC showed expression of the protein in all groups, while LPO was lost after rFGFR2b, pan-FGFR or MAPK inhibitor treatment (Fig. 6f). Taken together, these results show that differentiation of seromucous acinar cells requires FGFR2 signaling through the MAPK pathway.

### FGF7 and FGF10 can activate seromucous acinar transcriptional program

Activation of the acinar transcriptional program has therapeutic potential for regenerating exocrine secretory cells. Therefore, we asked whether the major ligands FGF7 and FGF10[41,42] could increase

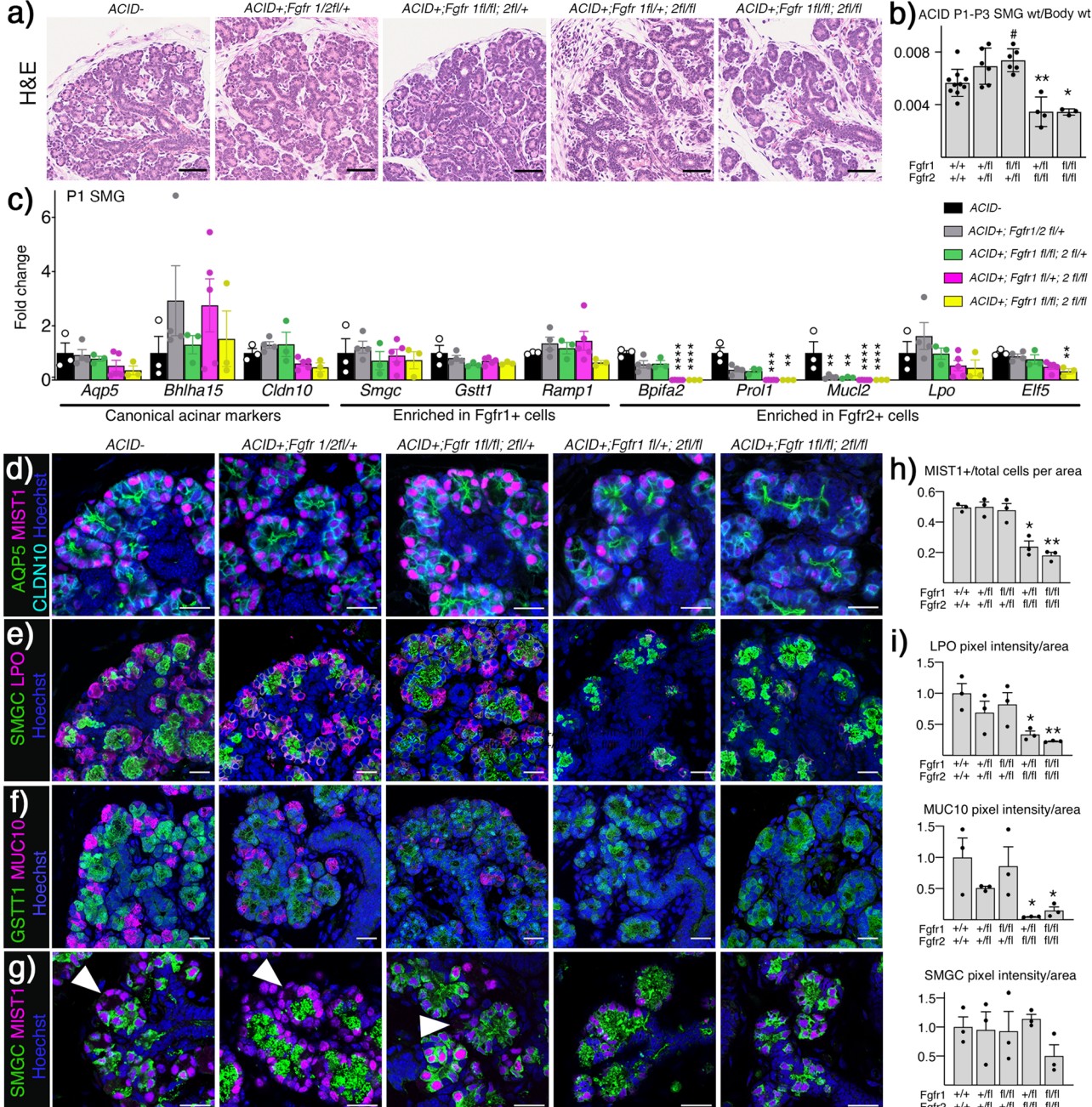

**Fig. 5 | SMG seromucous acinar cell differentiation is *Fgfr2*-dependent.**
**a** Deletion of *Fgfr2* led to acinar atrophy in P1 SMGs compared to either WT or *Fgfr1* deletion. Representative images stained with Hematoxylin and Eosin (H&E), scale bars: 50 μm. **b** *Fgfr2* deletion in acinar cells lead to decreased gland weight ratio compared to wildtype (Cre). Graph shows mean ratio of gland weight over body weight with SEM. One-way ANOVA with Dunnett's test for multiple comparisons to control (#*p* = 0.0165, **\**p* = 0.0068, \**p* = 0.0148), *n* = 10 for WT, *n* = 6 for *Fgfr1/2$^{fl/+}$* and *Fgfr1$^{fl/fl}$;2$^{fl/+}$*, *n* = 4 for *Fgfr1f$^{l/+}$;2$^{fl/fl}$* and *n* = 3 for *Fgfr1/2$^{fl/fl}$*. Source data are provided as a Source Data file. **c** Canonical acinar genes and genes enriched in *Fgfr1+* population were not affected by deletion of either *Fgfr1*, *Fgfr2* or both. Genes enriched in *Fgfr2+* cells decreased after deletion of *Fgfr2* or both receptors (n ≥ 3 for each genotype). One-way ANOVA with Dunnett's test for multiple comparisons to WT control (*Bpifa2* \*\*\*\**p* < 0.001, \*\*\**p* = 0.0003, *Prol1* \*\*\**p* = 0.0003, \*\*\**p* = 0.0011, *Mucl2* \*\**p* = 0.0033 and 0.0026, \*\*\*\**p* < 0.001, *Elf5* \*\**p* = 0.0093). *n* = 3 *ACID-*, *Fgfr1$^{fl/fl}$;2$^{fl/+}$* and *Fgfr1/2$^{fl/+}$*, n = 5 for *Fgfr1/2$^{fl/+}$* and *Fgfr1f$^{l/+}$;2$^{fl/fl}$*. Data are presented as mean values +/- SEM and source data are provided as a Source Data file. **d)** Representative images showing AQP5 (Aquaporin 5, green), MIST1 (magenta), CLDN10 (Claudin10, cyan) and Hoechst (blue) detected in P1 SMGs from all groups. Scale bars: 20 μm. **e** Deletion of either *Fgfr1*, *Fgfr2*, or both did not affect localization

pattern of SMGC (Submandibular gland protein C, green), while LPO (magenta) was not detected after *Fgfr2* deletion in P1 SMGs. Nuclei stained with Hoechst (blue). Representative images, scale bars: 20 μm. **f** Deletion of either *Fgfr1*, *Fgfr2*, or both did not affect localization pattern of GSTT1 (Gluthathione S-transferase theta 1, green), while MUC10 (Mucin10, magenta) was not detected after *Fgfr2* deletion in P1 SMGs. Nuclei stained with Hoechst (blue). Representative images, scale bars: 20 μm. **g** MIST1 (magenta), SMGC (Submandibular gland protein C, green) and Hoechst (blue) staining showed overlap in all MIST1 cells after *Fgfr2* deletion. Scale bar: 20 μm. Arrowheads indicating MIST1 + /SMGC- cells. Representative images, scale bars: 20 μm. **h** Quantification of IHC staining showed a decrease of MIST1+ cells per area after *Fgfr2* deletion (*n* = 3). One-way ANOVA with Dunnett's test for multiple comparisons to WT control (\**p* = 0.0007, \*\**p* = 0.0002). Data are presented as mean values±SEM. Source data are provided as a Source Data file. **i** Quantification of protein staining showed decrease in LPO and MUC10, while SMGC did not change following *Fgfr2* deletion (*n* = 3, normalized to WT). One-way ANOVA with Dunnett's test for multiple comparisons to WT control (LPO \**p* = 0.0229 \*\**p* = 0.0092, MUC10 \**p* = 0.0221 and 0.0396). Data are presented as mean values±SEM. Source data are provided as a Source Data file.

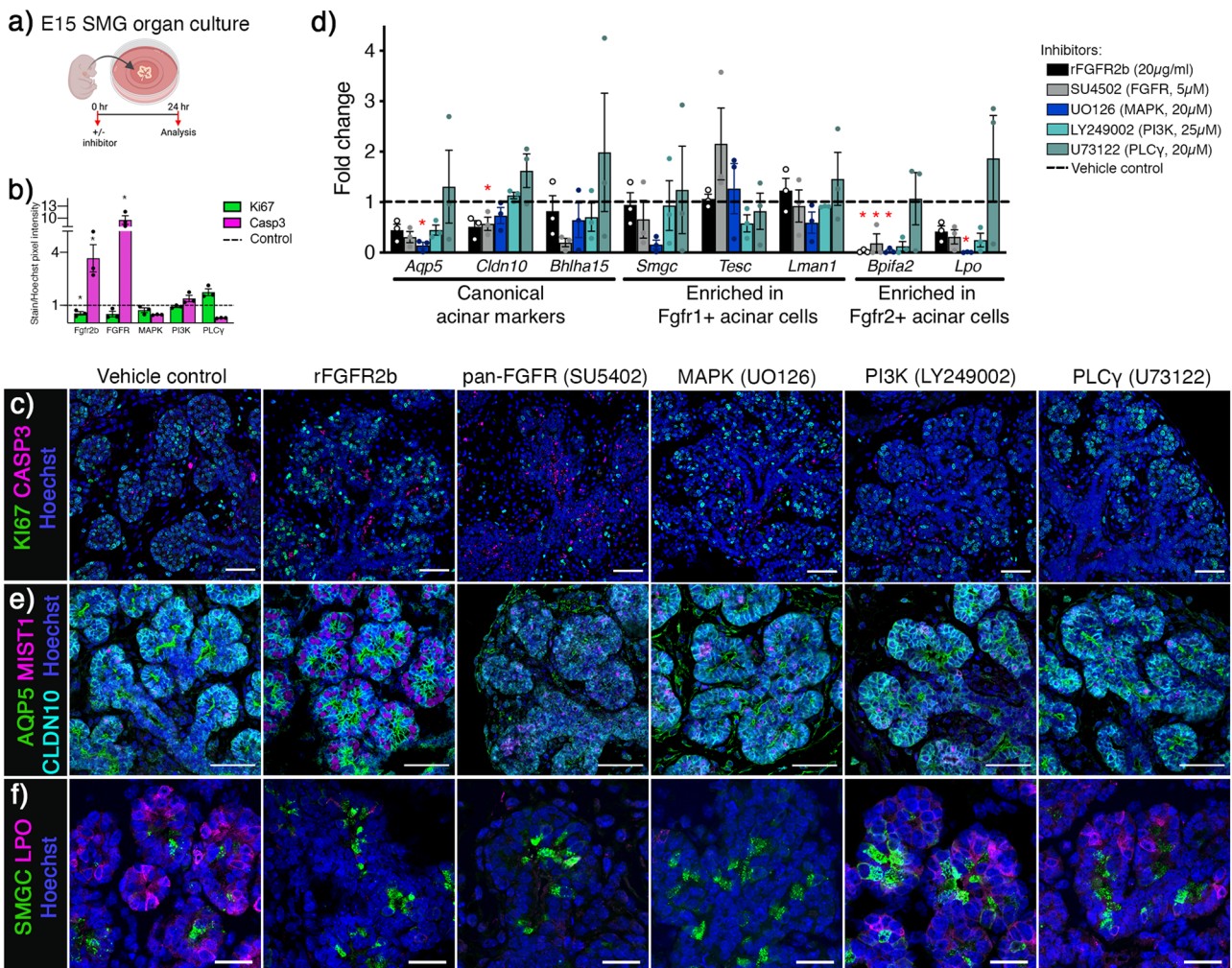

**Fig. 6 | *Fgfr2* is required for seromucous acinar differentiation through MAPK pathway. a** E15 SMGs were treated with inhibitors and cultured ex vivo for 24 h before analysis. Created using BioRender.com. **b** Quantification of Ki67 and Cleaved Caspase 3 (Casp3) staining showed increase in apoptosis after rFGFR2b (*$p$ = 0.0286) and pan-FGFR (*$p$ < 0.0001) inhibitors. Treatment with rFGFR2b decreased (*$p$ = 0.0114) while PLCγ inhibitor increased proliferation (*$p$ = 0.0177). MAPK or PI3K inhibitors showed no changes. One-way ANOVA with Dunnett's test for multiple comparisons to control and unpaired two-tailed t-test when comparing two groups (*$p$ < 0.05). $n$ = 3 for each treatment and staining and $n$ = 6 for control. Data are presented as mean values±SEM. Source data are provided as a Source Data file. **c** Representative images showing Ki67 (green) and Cleaved Caspase 3 (Casp3, magenta) in E15 SMG + 24 h. Nuclei stained with Hoechst (blue). Scale bar: 50 μm. **d** Gene expression of canonical acinar markers was decreased after FGFR inhibition. *Cldn10* was decreased after pan-FGFR inhibitor (SU5402, 5 μg/ml). *Aqp5* and *Lpo*

were decreased after MAPK inhibitor (UO126, 20 μg/ml), while *Bpifa2* was decreased after rFgfr2b (20 μg/ml), pan-FGFR- and MAPK-inhibitors. Genes enriched in the *Fgfr1*+ population did not change with the various inhibitors compared to vehicle control. One-way ANOVA with Dunnett's test for multiple comparisons to control and unpaired two-tailed t-test when comparing two groups (*Aqp5* *$p$ = 0.0031, *Cldn10* *$p$ = 0.0433, *Bpifa2* *$p$ = 0.0053, 0.0333 and 0.0074, *Lpo* *$p$ < 0.0001), $n$ = 3 for each treatment. Data are presented as mean values±SEM. Source data are provided as a Source Data file. **e** Acinar differentiation was evident in all groups shown through IHC of AQP5 (Aquaporin5, green), MIST1 (magenta) and CLDN10 (Claudin10, cyan). Nuclei stained with Hoechst (blue). Representative images, scale bars: 50 μm. **f** After 24 h, SMGC (green) could be detected in all groups, while LPO (magenta) was not detected after FGFR and MAPK inhibitor treatment. Nuclei stained with Hoechst (blue). Representative images, scale bars: 20 μm.

expression of *Fgfr2*-dependent secretory markers. Initial experiments treating E15 organ culture with exogenous FGF7 or FGF10 for 6 hrs showed no increases in acinar gene expression, likely due to the robust endogenous FGF production (Supplementary Fig. 5h). Therefore, we performed a gain-of-function experiment to test whether reduced seromucous differentiation after MAPK inhibition (UO126) could be restored or increased by exogenous FGF7 or FGF10. E15 glands were treated with MAPK inhibitor for 24 h before changing to fresh media containing either MAPK inhibitor, FGF7, FGF10 or vehicle control (Fig. 7a). Treatment with MAPK inhibitor for 48 h decreased *Aqp5*, *Bpifa2* and *Lpo* and washout with media for 24 h containing vehicle control rescued *Aqp5*, and partially rescued *Bpifa2* and *Lpo* expression (Fig. 7b). Addition of FGF7 further increased expression of both *Aqp5* and *Lpo* above the washout, while *Bpifa2* was restored to control levels

(Fig. 7b). Addition of FGF10 increased *Lpo* expression above the washout, but not *Aqp5* (Fig. 7b). Inhibitor treatment for 48 hrs decreased expression of *Smgc* and addition of ligands showed an increasing trend although not significant compared to washout alone (Fig. 7b). Addition of FGF7 or FGF10 for 24 h along with continued MAPK inhibitor treatment showed no changes in *Aqp5*, *Bpifa2*, or *Lpo* gene expression compared to inhibitor alone, confirming that the FGF7 or FGF10 stimulated changes act through the MAPK pathway (Fig. 7b). Accordingly, phosphorylated ERK was decreased following MAPK inhibitor and washout, while FGF7 or FGF10 treatment was comparable to control confirming FGFR2-dependent MAPK activation (Fig. 7c). Furthermore, immunostaining confirmed increased LPO protein expression following stimulation with either FGF7 or FGF10. In contrast, the decreased SMGC immunostaining with MAPK inhibitor

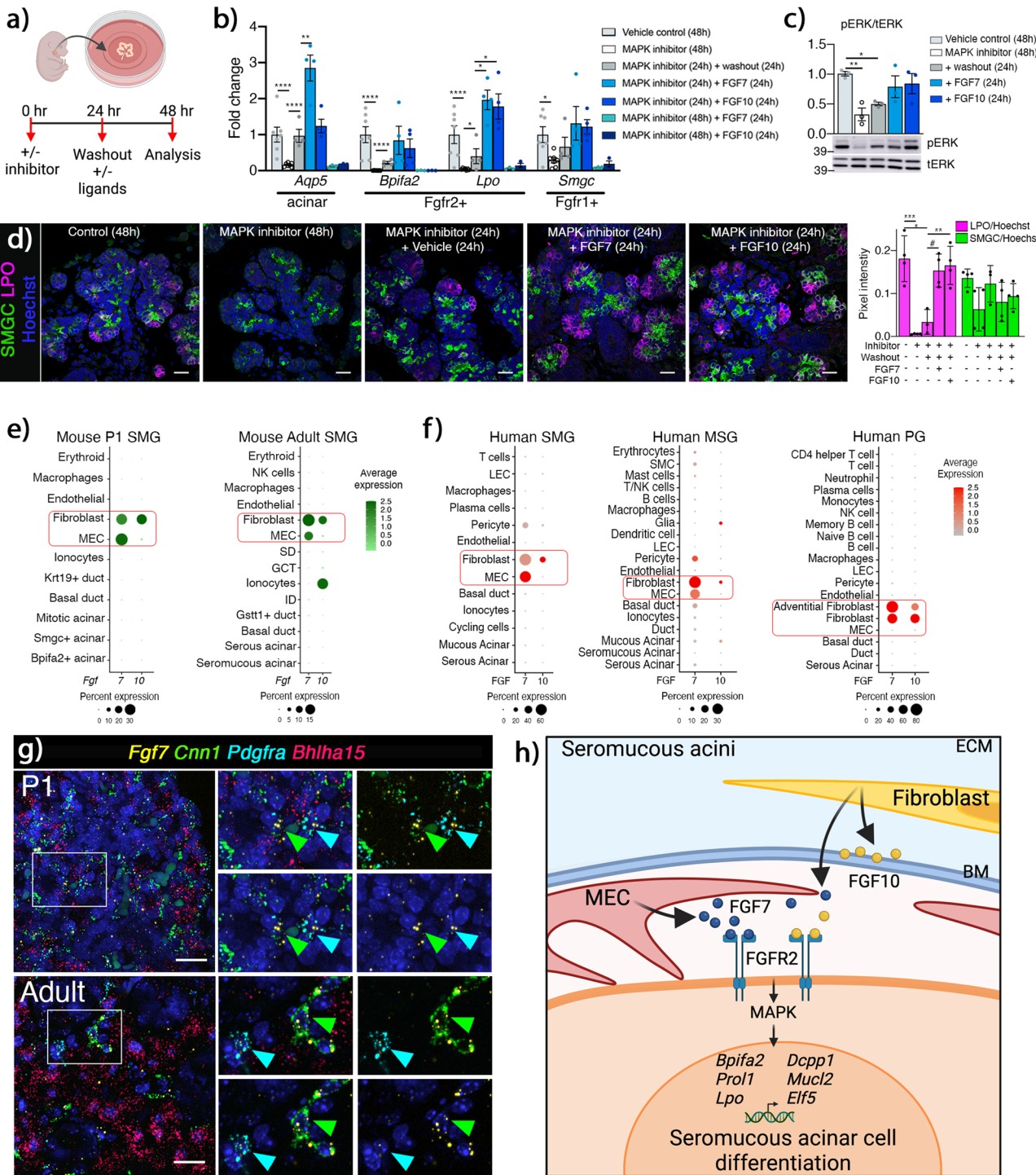

**Fig. 7 | Acinar and MEC crosstalk via FGF7-FGFR2 signaling can activate ser-omucous transcriptional program. a** Washout experimental timeline. Created using BioRender.com. **b** Genes reduced following MAPK inhibition could be par-tially rescued by inhibitor washout. *Aqp5* and *Lpo* were further increased by FGF7 and FGF10 treatment (*n* = 4). FGF7 or FGF10 had no effect in the presence of MAPK inhibitor (*n* = 3). One-way Anova with Dunnett's test for multiple comparisons (*Aqp5* ****p < 0.0001, **p = 0.0011, *Bpifa2* ****p < 0.0001, Lpo ****p < 0.0001, *p = 0.0264, 0.0144 and 0.0195, *Smgc* *p = 0.0402). Data are presented as mean values±SEM. Source data are provided as a Source Data file. **c** Western Blot showed decrease in pERK/tERK after MAPK inhibitor treatment and washout. Addition of FGF7 or FGF10 restored pERK/tERK comparable to control. One-way Anova with Dunnett's test for multiple comparisons, **p = 0.0096, *p = 0.0486), *n* = 3 for each treatment. Source data are provided as a Source Data file. **d** Representative images of SMGC (Sub-mandibular gland protein C, green) and LPO (Lactoperoxidase, magenta) in E15 + 48 h organ culture with inhibitor, washout and ligand treatment as indicated.

Scale ba = 20 μm. Quantification of LPO and SMGC IHC showed rescue of LPO protein by addition of FGF7 or FGF10. One-way Anova with Tukey test for multiple comparisons (***p = 0.0001, *p = 0.0014, #p = 0.0083, **p = 0.0038), *n* = 3 for washout group, *n* = 4 for all other groups. Data are presented as mean values±SEM and source data are provided as a Source Data file. **e** Dotplots showing expression of *Fgf7* and *Fgf10* in postnatal mouse SMG from scRNAseq data. **f** Dotplots showing FGF7 and FGF10 in human SMG, MSG and PG from scRNAseq. **g** Representative images of in situ hybridization with *Fgf7* (yellow), *Cnn1* (myoepithelial cells, green), *Pdgfra* (fibroblasts, cyan), and *Bhlha15* (acinar cells, red), show enrichment of *Fgf7* in myoepithelial and some in fibroblasts in both P1 and adult mouse SMG. Scale bar = 20 μm. **h** Crosstalk between acinar cells and MEC-derived FGF7 as well fibroblasts via FGF7/FGF10-FGFR2 drives seromucous acinar cell differentiation. MEC myoe-pithelial cell, ECM Extracellular matrix, BM Basement membrane. Created using BioRender.com.

was reversed after washout although ligand treatment did not show further increases (Fig. 7d). These results show that MAPK-dependent seromucous differentiation can be stimulated ex vivo by both FGF7 and FGF10.

Since activation of FGFR2 is critical for seromucous differentiation, we asked which cells are the source of the major ligands *Fgf7* and *Fgf10* in vivo. The acinar niche contains multiple cells, including MECs, nerves, blood vessels, immune cells, and fibroblasts, all producing signals needed for homeostatic acinar function. As expected, scRNA-seq showed that fibroblasts expressed both *Fgf7* and *Fgf10*, however surprisingly, *Fgf7* was also predominantly expressed in MECs, while *Fgf10* is expressed in mature ductal ionocytes and fibroblasts (Fig. 7E), as recently reported[43]. Interestingly, human SGs showed a similar expression pattern, with FGF10 in fibroblasts and FGF7 expressed in MECs of SMG and MSG, but not detected by scRNAseq in PG MECs (Fig. 7f). We confirmed the novel expression of *Fgf7* in MECs and fibroblasts in both P1 and adult SMGs through in situ hybridization (Fig. 7f). In vivo, MECs directly contact and wrap around the acini, and this acinar complex is surrounded by the basement membrane, whereas the stromal fibroblasts that produce FGF10 are separated from acini by the basement membrane. Taken together, we hypothesize that FGF7 secreted from MECs that are in direct contact to acini provide critical niche signal for seromucous acinar differentiation. Taken together, we have shown that seromucous acinar cell differentiation is dependent on FGFR2-MAPK signaling via FGF7 and FGF10 (Fig. 7h).

## Discussion

We have previously defined roles for FGFR signaling during SMG branching morphogenesis and endbud expansion[34,44]. Here, we used genetic tools to conditionally delete FGFRs in vivo, identifying essential and cell-specific roles for FGFR signaling during gland initiation, duct homeostasis and seromucous and serous acinar differentiation. Leveraging scRNAseq data to map FGFRs to specific cell types during both embryonic SG and postnatal development, allowed for precise predictions of lineage behavior which were confirmed by the different outcomes of deleting FGFRs in either the entire ectodoerm (Crect), basal epithelium (Krt14Cre), ductal lineage (Krt5Cre) or acinar lineage (ACIDCre). As predicted, deleting FGFR2 in ectodermal and epithelial lineage led to salivary agenesis, confirming the requirement of FGFRs for early SG development. Surprisingly, loss of epithelial FGFR1 in the absence of FGFR2 increased the severity of mandibular and maxillar hypoplasia in Crect + ;*Fgfr1*$^{fl/fl}$;*Fgfr2*$^{fl/fl}$ embryos, suggesting a specific and additional functional role for FGFR1 in craniofacial development in the absence of FGFR2. We identified essential FGFR functions in progenitor lineage contribution and maturation of ducts. Importantly, we discovered that FGFR2-MAPK signaling was critical for seromucous and serous acinar cell differentiation in the SMG and SLG, respectively, while FGFR1 was dispensable. We propose a working model, where FGF7, expressed by MECs in contact with acinar cells, acitivates the FGFR2-dependent seromucous transcriptional program to increase saliva secretion (Fig. 7h).

FGF10-FGFR2b signaling is required for epithelial homeostasis, self-renewal and regeneration in basal cell progenitors in several tissues such as airways, prostate, and mammary glands[45–48]. In SGs, the current model is that basal cells are lineage restricted progenitors from the onset of differentiation, contributing to both duct homeostasis and regeneration[29,49,50]. Irradiation injury induces their plasticity, which leads to limited long-term regeneration of acinar cells[49]. Still, signaling factors required to drive progenitor function in SG basal cells are not well-characterized. We recently reported that salivary ionocytes express FGF10 and predicted a potential interaction with FGFRs in adult basal cells[43]. Here, we show that FGFR deletion in basal cells leads to an abnormal duct phenotype in adult glands. Stage-specific FGFR deletion in adult basal cells reduced their lineage contribution,

highlighting a critical role for FGF signaling in postnatal duct progenitor function similar to other stratified epithelia. We propose ionocyte-basal cell crosstalk via FGF10-FGFR1/2 as a potential driver of postnatal duct homeostasis. Whether FGFR1 and FGFR2 have differential roles and are required for injury induced plasticity in SG basal cells remains to be determined.

A major challenge for acinar regeneration is the identification of specific signals inducing the functional secretory cell type. Consequently, several developmental studies have focused on signaling pathways and transcription factors critical for progression and timing of endbud development and specification into acinar cells. In SGs, formation of the initial bud with SOX9+ and SOX10+ distal cells is driven by FGF10-FGFR2b signaling[26,36]. By E14, combined FGFR2b and KIT signaling specifically expands the SOX10+ distal endbud progenitors and amplifies FGFR2b-dependent transcription[44]. Recently, NRG1-ERBB3 neuronal-epithelial crosstalk was implicated in acinar specification during development, driving onset expression of acinar markers such as A*qp5* and *Bpifa2*[51]. Here, we show the "dose-dependent" requirement of FGFR2b signaling via MAPK pathway in acinar cells driving the secretory differentiation of seromucous cells in the SMG. We genetically deleted FGFRs in cells expressing *Aqp5*, implying they were committed towards an acinar fate. Accordingly, acinar cell fate was not prevented as canonical acinar markers were still present after both FGFR1 and FGFR2 deletion, although acinar morphology and secretory protein production were severely disrupted. It is likely that a combination of niche factors such as EGFR and FGFR ligands may be required to initiate acinar cell fate and then drive seromucous and serous differentiation, respectively.

During embryonic development, mesenchymal cells provide the FGFR ligands FGF10 and FGF7 required for epithelial development[14]. Here, we found that MECs are one of the major sources of *Fgf7* in postnatal SGs. Considering the direct contact between MECs and acinar cells within the basement membrane, identifies them as critical niche cells that may drive seromucous and serous acinar secretory transcriptional profiles. Further, we show expression of FGFR2 and FGF7 in adult acinar and MECs respectively, suggesting continued acinar-MEC crosstalk via FGF7-FGFR2 is important for seromucous differentiation and function, which may stimulate secretion in adult glands. This is supported by the observation of increased saliva flow, evidenced by drooling mice with wet necks and chests, when FGF7 was overexpressed in basal cells and MECs in vivo[52]. Further, Palifermin which is a truncated version of FGF7 and an FDA-approved drug used to treat mucositis in patients receiving chemo- and/or radiation-therapy[53], caused an increase in acinar area in histology sections and increased acinar proliferation, when injected into mouse SGs before irradiation[54]. Whether such treatment increases the seromucous and serous acinar cells remains to be determined, and further work including deletion of *Fgf7* in MECs using a recently reported FGF7$^{flox}$[55] may support our working model for the critical role of niche cell signals.

Here, we have established cell-specific roles for FGFR signaling during epithelial development, duct homeostasis and acinar differentiation. It is not clear whether disruption of FGFR2 signaling is directly involved in acinar pathogenesis in a setting of gland dysfunction post irradiation. Also, additional factors are likely involved in acinar regeneration post-irradiation damage, which was increased by neuronal stimulation that induced acinar cell proliferation[56]. It was also recently shown that irradiation of human SGs leads to MEC-specific upregulation of neurotrophin receptors, which potentially hinder acinar regeneration[57]. Still, gene therapy with retroductal infusion of an adenovirus expressing FGF7 restores salivary flow in irradiation-induced salivary hypofunction[58]. Similarly, intraglandular injections of FGF7 has protective effects of murine SGs in vivo, as well as human SG cells in vitro, following irradiation- or radioiodine-induced hypofunction[54,59,60]. The proposed mechanism in these reports

include protecting acinar cells from apoptosis and stimulating the basal duct progenitors to differentiate into acinar cells. The protective effects of FGF7 were proposed to include maintaining acinar, MEC and endothelial markers as well as saliva flow[59,60]. Altogether, these reports suggest an important role for FGF7-FGFR2 signaling in several cell types after irradiation damage and highlights potential for increasing acinar differentiation during tissue regenerative strategies.

Our findings provide in vivo evidence that FGFR2 signaling will be vital for regenerative therapies that require seromucous and serous acinar differentiation as part of a fully functional SG. Any regenerative therapy will need to take into consideration multiple factors that directly drive acinar specification and differentiation, as well as MEC-specific factors that influence acinar regeneration. We propose a central role for FGF7-FGFR2 signaling allowing for differentiation and maturation of specific acinar cell types, acinus morphology, as well as duct progenitor function.

## Methods

### Materials availability

No specific new materials were generated in this work, see Supplementary Table 1 for resources.

### Mouse strains

All mouse strains used have been previously described and include *Crect*, *Krt14Cre*, *Krt5Cre*, *KrtSrtTA;tet-Cre*, *ACID*, *Fgfr1*fl/fl, and *Fgfr2*fl/fl alleles (see Supplementary Table 1). Reporter strains mTmG (*Gt(ROSA) 26Sor*tm4(ACTB-tdTomato,-EGFP)Luo/J, The Jackson Laboratory) and tdTomato (B6.Cg-*Gt(ROSA)26Sor*tm9(CAG-tdTomato)Hze/J, The Jackson Laboratory) in addition to timed pregnant ICR (CD-1®) females (Envigo) were used. Mice carrying *Fgfr1*fl/fl and *Fgfr2*fl/fl were mated and subsequently maintained as a double floxed strain. To generate the *Fgfr1* fl/fl;*Fgfr2*fl/fl embryos used in the study, timed mating was set up, and the morning a plug was detected was considered day 0. Mice were kept in a 14 h on/ 10 hr off light/dark cycle at 74–78 F with 50–70% humidity. Genotyping was performed using standard protocols (See Supplementary Table 1 for specific primers). Due to the known sexual dimorphism in adult mouse SMGs, differing outcomes between sexes was considered when using adult mice for experiments. Sex was not considered in experiments using embryonic or neonatal glands since no sexual dimorphism is present during development. All experiments were approved by the NIH Animal Care and Use Committee.

### Doxycycline treatment

Adult mice (6–8 weeks old, males and females) were fed Doxycycline diet (5001 C w/6000ppm, Animal Specialties and Provision, PA, USA) ad libitum for 4 days (day 0), before changing the food back to standard diet. Tissues were harvested for analysis at indicated timepoints. For induction during embryonic development, females were fed doxycycline during pregnancy.

### Organ culture explants

SMGs were dissected from ICR E15 embryos and placed on Whatman Nuclepore Track-etch filters (13 mm, 0.1 µm pore size; VWR, Buffalo Grove, IL) with 200 µL fresh DMEM/F12 media (from Thermo Fischer Scientific) containing 1% Penicillin-Streptomycin, Transferrin (150 µg/ml, from Thermo Fischer Scientific) and Vitamin C (50 µg/ml, from Sigma Aldrich). Two to three glands from separate embryos were placed per filter and allowed to set for 2 h before randomly selected for addition of inhibitors targeting FGFR signaling and its downstream effectors. Mouse recombinant Fgfr2b Fc Chimera (R&D Systems, 20 µg/ml) was run along with its vehicle control (BSA). Inhibitor stocks of SU5402 (Millipore Sigma), UO126 (Millipore Sigma), U73122 (Millipore Sigma), and Ly249002 (Biotechne) were resuspended in DMSO (Sigma Aldrich) as per vendors instructions. Final inhibitor dosages of 5, 20, 20, and 25 µM respectively were used, and experimental groups were run along with

equivalent vehicle (DMSO) as control. Each treatment group was repeated using 3 independent litters ($n = 3$, for each treatment) and each litter had control groups ($n = 3$ for BSA, $n = 12$ for DMSO control). SMGs were cultured at 37 °C in a humidified 5% $CO_2$/95% air atmosphere for 24 h (unless otherwise noted). RNA was isolated from homogenized glands before qPCR gene expression analysis as described below.

Organ culture experiments with washout and ligand treatments were cultured under the same conditions as described above. Per dish, two glands from separate embryos were randomly selected for treatment group. Each treatment was replicated using 3 independent litters (n = 3 for each treatment group). Here, glands were treated with UO126 (20 µM) or vehicle (DMSO) for 24hrs before washout with fresh media 3×5 minutes. Glands were then incubated again for 24 h with either UO126, DMSO, FGF7, FGF10 or UO126 combined with either FGF7 or FGF10. (All recombinant FGFs were from R&D Systems). Final dosages were 500 ng for FGF7 and FGF10 (volume of 10 µL).

### Real time qPCR

RNA was isolated using either RNAqueous-4PCR total RNA isolation kit or RNAqueous-PCR micro kit with DNase treatment (Both from ThermoFisher). cDNA was made using the iScript cDNA Synthesis Kit (Bio-Rad) and 1 ng was amplified with 40 cycles of 95 °C for 10 s and 62 °C for 30 s. All qPCR was run using a C1000 Touch Thermal Cycler and CFX96 Real-Time System connected with the Bio-Rad CFX Maestro 2.3 Software (All from Bio-Rad). Gene expression was normalized to the house-keeping gene, *Rps29* and the control groups. Fold change was calculated for each sample and normalized to control. All graphs show mean fold change +/- SEM. Amplification of a single product was confirmed by melt curve analysis and all reactions were run in duplicate. Beacon Designer™ software (PREMIER Biosoft) was used to design primers for this study. See supplementary Table 1 for specific sequences.

### Immunohistochemistry

For frozen sections, tissues or whole embryos were fixed in 2% paraformaldehyde (Electron Microscopy Sciences, #15700) overnight (ON), then stored in PBS. Tissues were dehydrated with increasing sucrose concentrations, 15%, 30%, and then 1:1 30% sucrose: OCT™, until equilibrated, then embedded in Tissue-Tek® OCT™ compound (Sakura) at 0°C. Frozen sections of 10 µm or 50 µm were cut and placed on Superfrost Plus glass slides (Thermo Fisher Scientific). After blocking with 10% Normal Donkey Serum for 1 hr at room temperature (RT), sections were incubated with primary antibody ON at 4 °C (SOX10 1:100, Santa Cruz Biotechnology, sc-17342) followed by secondary antibody incubation for 1 hr at RT (Alexa Fluor® 647 AffiniPure F(ab')₂ Fragment Donkey Anti-Goat IgG (H + L), Jackson Immunoresearch Laboratories, #705-606-147). Cover slips were mounted using Fluoro-gel II, with DAPI (#17958-50, Electron Microscopy Sciences) and set overnight at room temperature. Sections were washed in PBS between each step.

For paraffin sections, tissues were fixed in 4% paraformaldehyde (Electron Microscopy Sciences, #15700) ON and transferred to 70% ethanol until day of embedding. Tissues were embedded in paraffin, and sections cut following standard procedure (Histoserv Inc., Germantown, MD). For staining, sections were deparaffinized in Xylene substitute (Sigma Aldrich, A5597-1GAL) followed by hydration through a graded series of ethanol (100-95-70%) to $H_2O$. The slides underwent heat-induced antigen retrieval using a Tris-EDTA pH 9 buffer (TrisBase T1378, EDTA E9884 from Millipore Sigma) using a pressure cooker. Slides were allowed to cool down and then blocked for 1 hr with either normal donkey serum (Jackson Immunoresearch Laboratories) or M.O.M block (Mouse on Mouse Immunodetection Kit, Vector Laboratories, BMK-2202). Primary antibodies were incubated ON at 4 °C followed by 1 hr incubation in secondary antibody. Primary antibodies used were E-cadherin (1:200, BD Biosciences, #610182), GFP

(1:500, Abcam ab13970), SMGC (1:200, Lifespan Bioscience, LS-C154825), LPO (1:200, Thermo Fisher Scientific, PA1-46353), GSTT1 (1:100, Lifespan Bioscience # LS-B10781), MUC10/PROL1 (1:200, Everest, EB10617), Ki67 (1:200, BD Pharmigen, 550609) and Cleaved Caspase 3(1:100, Cell Signaling, #9664 S). Secondary antibodies used were Alexa Fluor® 488 AffiniPure F(ab')$_2$ Fragment Donkey Anti-Goat IgG (H + L) (1:250, #705-546-147), Alexa Fluor® 647 AffiniPure F(ab')$_2$ Fragment Donkey Anti-Goat IgG (H + L) (1:250, #705-606-147), Cy™3 AffiniPure F(ab')$_2$ Fragment Donkey Anti-Rabbit IgG (H + L) (1:250, #711-166-152) all from Jackson Immunoresearch Laboratories. Sequential staining was done if several primary antibodies were used. Nuclear staining with Hoechst (Thermo Fisher Scientific, H3570) was done and coverslips were mounted using Thermo Scientific™ Shandon™ Immu-Mount™ (#9990402). Slides were washed in PBS between each step.

Co-staining done with same species antibodies were performed using a multiplex method. Paraffin sections were deparaffinized as described above before antigen retrieval in a microwave (2:30 mins at 700 W power followed by 5 minutes at 300 W power) in Tris-EDTA pH 9.00 buffer before 1 hr cool down. Slides were incubated 10 minutes in Bloxall® Endogenous Blocking Solution (Vector Laboratories, SP-6000) before primary antibody incubation at 4 °C ON. Primary antibodies used were MIST1 (1:500, Cell Signaling, #14896), AQP5 (1:700, Alomone labs, AQP-005) and CLAUDIN10 (1:1000, Thermo Fisher Scientific, #38-8400). Slides were rinsed in PBS and incubated 30 minutes at RT in Rabbit IgG VisUCyte HRP Polymer Antibody (R&D Systems, VC003-025). Sections were rinsed in PBS and treated with pre-incubation buffer (0.1 M Boric Acid, 2 M Salt Chloride, 0.2 mg/mL 4-Iodophenylboronic acid) for 10 minutes before 10 minutes with Tyramide reagent (diluted in buffer: 0.1 M Boric Acid, 2 M Salt Chloride, 0.2 mg/ml 4-Iodophenylboronic acid and 0.003% H2O2). Tyramide reagents used were Alexa Fluor™ 488 Tyramide Reagent (1:200, B40953), Alexa Fluor™ 594 Tyramide Reagent (1:200, B40957) and Alexa Fluor™ 647 Tyramide Reagent (1:100, B40958). Antigen retrieval step and staining steps were then repeated until all antibodies had been stained. Nuclear staining and cover slips were mounted as described above. All images were captured using a Nikon A1R + MP microscope (40x or 60x oil objectives) and the Confocal NIS-Elements Package (Nikon instruments).

Hematoxylin and Eosin (H&E) staining of paraffin sections were performed by Histoserv Inc (Germantown, MD). Slides were scanned with a S60 NanoZoomer Digital slide scanner (Hamamatsu) and images were exported using the NDP.view 2 software (Hamamatsu).

### In situ Hybridization
Using RNAse free solutions, SMGs from P1 and adult ICR mice were isolated (n = 3), washed in PBS and fixed in 4% PFA before paraffin embedding. Sections were sent to Advanced Cell Diagnostics (ACD) for in situ hybridization for *Fgfr1, Fgfr2, Bhlha15, Krt5, Cnn1, Pdgfra, and Fgf7*. Specific probe sequences are proprietary and generated with RNAscope® technology by ACD. Images were taken using a Dragonfly Spinning Disk system (60x objective).

### Western Blot
After treatment, E15 SMG organ culture samples were sonicated in RIPA Buffer with protease and phosphatase inhibitor cocktail (both from Thermo Fisher Scientific) and the total protein concentration was measured using a BCA Protein Assay Kit (Thermo Fisher Scientific). The samples (20 μg) were then heated at 95 °C for 10 min in NuPage™ LDS Sample Buffer and NuPage™ Sample Reducing Agent and run on NuPage™ 4–12%, Bis-Tris mini protein gels (all from Thermo Fisher Scientific). Proteins were transferred to PVDF membranes using iBlot™ 2 Transfer Stacks and iBlot 2 Dry Blotting System (Thermo Fisher Scientific). Blots were blocked with 5% BSA in 0.1% TBS-Tween (10× TBS

and 0.1% Tween-20) for 1 h at room temperature before incubation in primary antibodies ON at 4 °C (phospho-p44/42 MAPK, 1:2000, #4370 S and p44/42 MAPK, 1:2000, #9102 S, both from Cell Signaling). All blots were incubated in Anti-rabbit HRP-linked secondary antibody (1:10,000, Cell Signaling, 7074) and developed using the SuperSignal West Dura Chemiluminescent Extended Duration Substate (Thermo Fisher Scientific). Blots were imaged on a GE Amersham Imager AI680 using automatic exposure settings and analyzed using FIJI software. Representative full scan blot is available in the source data file and supplementary information.

### Computational analysis
All scRNAseq data was analyzed using R and R studio (https://rstudio.com/) and Seurat V4. Using previously annotated mouse SMG scRNAseq data, epithelial populations were computationally separated SEURAT's subset function. Epithelial subsets from E12 were re-normalized and scaled to generate new SEURAT objects. E16 and P1 acinar subsets were integrated through standard pipeline to generate a new SEURAT object with mouse acinar cells. All statistics for the computational analyses to determine significant markers were performed using the default pipeline statistical test in SEURAT. This analysis is based on non-parametric Wilcoxon rank sum test and adjusted p-values of <0.05 were chosen as a measure of significance. Human MSG datasets (GSE180544 and https://www.covid19cellatlas.org/) were integrated and previous annotation from the two datasets were harmonized and used to identify cell populations. Human SMG and PG datasets were imported (GSE201333) and the previous annotations were used to identify populations.

### Quantification of histological images
Images were taken from random areas of H&E-stained *Krt5Cre; Fgfr1$^{fl/fl}$; Fgfr2$^{fl/fl}$* slides as described above. Duct/total area ratio was calculated by measuring the area of duct and total area using FIJI[8]. For quantification of fluorescent images, imaging was performed using a 40× objective on a Nikon A1R + MP microscope using resonant scanning method. From *Krt5rtTA; tetCre;Fgfr1$^{fl}$;Fgfr2$^{fl}$;mTmG* sections 10 random areas were selected (5 μm thick confocal stacks with 0.5 μm steps) from at least 3 different animals (n = 3). The GFP and Hoechst expression intensity was measured by calculating the integrated density value of the samples histogram resulting from the maximum intensity projection. Quantification of the 10 points were averaged and GFP was normalized to Hoechst staining. Results were then averaged by biological replicates. Quantification of MIST1+ cells per total cell number was done by imaging 3–4 random areas per gland (n = 3). MIST1 + and total nuclei (Hoechst) per area was counted using FIJI. For quantification of Ki67, Cleaved Caspase3, SMGC, LPO, and MUC10 staining, 2–4 confocal stacks (5 μm stacks, 0.5 μm steps) from 3 glands (n = 3) for each treatment and 6 glands (n = 6) for DMSO control were processed using the denoise function of the Confocal NIS-Elements Package prior to quantification. Reduction of background was done either by using the subtract background function or threshold with stack histogram, and integrated density was measured on maximum projections and normalized to either Hoechst or area as well as WT control as indicated (using FIJI). All quantification measurements of confocal imaging were performed blinded.

### Statistics and reproducibility
To compare more than two experimental groups, we performed one-way ANOVA with post hoc Dunnett's or Tukey test (as indicated) and for comparison of two data sets, the student's t-test with two-tailed tests and equal variance was used to calculate p-values. Graphs show mean ± SEM for each group from three or more biological replicates (as indicated). Images of IHC are representative and each staining was repeated for at least 3 biological replicates showing similar results.

**Reporting summary**

Further information on research design is available in the Nature Portfolio Reporting Summary linked to this article.

## Data availability

All scRNAseq used in this study are from previously deposited data (supplementary Table 1). Murine SMG at multiple developmental stages is from GSE150327[30]. Human minor salivary glands scRNAseq datasets are from GSE180544[32] and through a website portal https://www.covid19cellatlas.org/[33], while human SMG and PG scRNAseq is from GSE201333[31].

Specific values used to generate graphs in this paper are provided in the Source data file. Source data are provided with this paper.

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

## Acknowledgements

We thank Elsa Berenstein, NIDCR imaging core (ZIC DE000750-01) and NIDCR veterinary resource core (ZIC DE000740-05) for excellent technical assistance. We thank Drs. Vaishali Patel, Ana Costa da Silva and Alejandro M. Chibly for valuable feedback and discussions. This research was supported by the Intramural Research Program of the National Institute of Dental and Craniofacial Research at the National Institutes of Health, Grant DE000722 to M.P.H.

## Author contributions

Conceptualization; M.H.A., J.M.S., W.M.K. and M.P.H. Investigation; M.H.A., J.M.S., W.M.K., C.U.V. and J.T.D. Data analysis; M.H.A., J.M.S., W.M.K. Writing Original Draft, Review & Editing; M.H.A., J.M.S., W.M.K., C.U.V., J.T.D., S.W. and M.P.H. Resources and funding acquisition; M.P.H. Supervision; M.P.H.

## Funding

## Competing interests

The authors declare no competing interests.
