## [Peer Review File · Nature Communications]

REVIEWER COMMENTS

Reviewer #1 (Remarks to the Author):

In this study, Aure et al use multiple in vivo transgenic mouse lines and ex vivo organ culture to define the role of Fgfr1 and Fgfr2 during multiple stages of salivary gland organogenesis including early development, acinar specification, and acinar and duct differentiation. So far, research investigating these receptors has been restricted to early SG development, and this study provides a new perspective on FGF signaling during later stages of SG establishment. Using single cell approaches, the authors build off previous work characterizing gene expression to differentiation time points of multiple cell types. Findings delineating enrichment of Fgfr1 and Fgfr2 in subpopulations of acinar cells during early postnatal development are extremely interesting and experiments suggest an important role for these receptors in SG homeostasis. Results here show great promise, however at this time, areas of the manuscript require major revisions as statements and conclusions are not fully supported by experimental evidence. Edits to the manuscript are also recommended to create a more coherent story and emphasize the importance of the findings. Comments have been discussed below and have been organized into sections of the manuscript for clarity.

Introduction:

1. The introduction is lacking an important explanation of the different stages of gland development, specification, and cellular differentiation, with corresponding timepoints. Without this, it is difficult to understand why the timepoints used at all stages of this study were chosen.
2. Background on the need to study salivary gland development and biology as a therapeutic strategy for multiple pathologies is very vague. Further explanation is needed.
3. The line “Despite the central role of acinar cells,....mucus containing saliva compared with watery serous saliva” is not scientifically sound. Both mucous and serous fluids contain water and mucins. It is suggested to remove this statement and replace with something more appropriate to describe different cell types.
4. Further introduction to FGFRs and their ligands is required. The authors only discuss Fgfr1b and Fgfr2b, yet both Fgfr1 and Fgfr2 have two major isoforms (IIIb and IIIc). These have not been addressed. Furthermore, the authors use Fgfr1b and Fgfr2b terminology throughout the manuscript yet their approach using single cell seq and mouse genetics are not isoform specific, therefore they cannot draw conclusions on IIIb isoforms alone. This is even more important due to previous work defining the Fgfr2c isoform to be essential for SMG development (See Jaskoll et al., 2002). The authors define their results

as Fgfr2b specific, yet they cannot rule out the role of Fgfr2c. Terminology should be adjusted to discuss Fgfr1 and Fgfr2 only.

5. The statement “Studies using murine models have established that both FGFR1b and FGFR2b are expressed in the epithelium during embryonic development and the ligands, FGF10 and FGF7, are expressed in the mesenchyme” is not referenced.

Figure 1:

6. It is difficult to understand the explanation of Fgfr1 and Fgfr2 expression during embryonic timepoints with relation to Figure 1 A. The dot plot is clear, however the text gives a general overview without discussing individual timepoints shown in the Figure. The authors explain that 88% of cells in the endbud cluster express both Fgfr1b and Fgfr2b in embryonic glands. Is this in all timepoints combined? Percent of expression in dot plots are also heterogenous for both Fgfr1 and Fgfr2 yet both receptors are described to have the same expression. Please review and provide a more detailed analysis of Fgfr1 and Fgfr2 expression of each time point during embryonic and postnatal/adult stages.

7. This group previously defined numerous end bud clusters within these embryonic timepoints, yet one endbud cluster is described here. Please explain if all clusters were collected and analyzed as one and justify the reasoning for this. “Acinar populations previously described” have not been explained.

8. Figure 1 B shows 3 samples for P1 and 2 adult for acinar cells. This has not been explained.

9. Authors have defined that they will focus their analysis on Fgfr1 and Fgfr2 but it has not been explained why they only investigate transcript expression of P1 and adult in Figure 1 C-D when they have just described embryonic expression.

10. Figure 1 B shows Fgfr1 expression to be more widespread in the Krt5+ population at P1, yet FISH shows Fgfr2 transcripts (red) to be a lot more numerous than Fgfr1 (white) transcripts and Fgfr1 expression looks very minimal. Please explain and define how many samples were used to validate transcript expression.

11. The authors vaguely describe “expression pattern was similar to that of adult human SMG, MSG, and PG” yet dot plots provided in supplementary show differing patterns and also the introduction of new cell types that have not previously been defined (e.g. ionocytes). Please describe all data in manuscript that is shown in figures.

Figure 2

12. The authors show beautiful images of phenotypes in both the *Tfap2a*-Cre and *Krt14*Cre mouse lines, yet it has not been explained why two separate Cre drivers have been used. The reasoning and benefits of this need to be described.

13. E12 buds look like they are at a younger timepoint in the *Cre*-Cre (Figure 2 E) than the *Krt14*Cre (F) and SOX10 expression has not been confirmed in these samples. It is recommended to stain E12 *Cre*Cre samples for SOX10 to determine if acinar progenitor commitment occurs as in the WT samples.

14. Can the authors explain why they believe there is a differing phenotype between these two Cre lines? Single panels of stains are recommended as it is difficult to see SOX10 expression with green and red overlapping.

15. It is stated that “Both cre models generated embryos with gross morphological changes that were non-viable due to lack of organ development upon *Fgfr2b* deletion.” Was this confirmed by the authors or assumed based on previous global KO models? It is generally understood that the *Fgfr2b* KO is non-viable due to lung agenesis, yet the lung is endoderm derived. It would therefore be expected that lung development would not be affected in the *Cre* Cre and non-viable embryos are caused by cranial abnormalities driven by loss of global *Fgfr2* and further emphasize the importance of *Fgfr2c*.

Figure 3

16. It is not clear why the authors have chosen two separate *Krt5* cre mice. A constitutive *Krt5* Cre is used and only adult female mice are analyzed, and then an inducible *Krt5*Cre is used to delete *Fgfr* alleles during early development. It is interesting that the authors did not use the constitutive cre to investigate embryo development, as results shown in Figure 3 A could be driven by an embryonic or adult phenotype. Please explain.

17. Authors define ductal hypoplasia in glandular tissue sections, yet an increase in the acinar population could also lead to fewer visible ducts with this analysis method. This needs to be investigated further before ductal hypoplasia can be determined. Quantitative gene and/or protein expression analysis of ductal and acinar markers is encouraged to define this.

18. It appears by the images provided in Figure 3 that male mice have a significant ductal phenotype with 90 day chase of inducible *Krt5* driven deletion of *Fgfr1* and *Fgfr2*. Mis-organization of ductal cells and collapse of the luminal network seem apparent by ECAD and nuclear staining, yet the authors have not commented on this. Do the authors believe FGFRs have a more significant role in male GCT homeostasis vs female? Single cell analysis in adult mice was carried out on females only, so it cannot be

confirmed if FGFRs have the same cellular expression in males as females. FISH of *Fgfr1* and *Fgfr2* in adult male mice is advised to support the inclusion of male glands in this figure.

19. Lineage tracing is very informative of the role of *Fgfrs* in progenitor cells replenishment of the GCT cells. The quantification method used to calculate GFP+ cells is questionable, as different protein content of cells can contribute to different GFP cell membrane intensity (or intracellular intensity as observed in male GCTs). GFP+ cells as a percentage of all GCT cells should be calculated for each condition to determine lineage of *Krt5* progenitors.

20. Figure 3 D schematic is confusing, as it outlines the hypothesis and not a result. Schematics should be used to define the result of an experiment so it is recommend to adjust this and place at the end of Figure 3 to confirm discovery.

Figure 4

21. The reason for combining E16 and P1 datasets to analyze acinar cells needs to be more clearly described (e.g. previously defined stages of acinar differentiation).

22. The authors highlight a population of cells that were eliminated based on ambiguous expression of markers of multiple cell lineages, however the group of genes listed are known molecular signatures of myoepithelial cells and may indicate differentiating MECs from end bud cells. This population may hold significance due to high FGFR expression in the MEC population (as described earlier in Figure 1) and should be further investigated and/or exclusion needs to be further justified. Please see Chatzeli et al., 2023 Developmental Cell with regards to early MEC differentiation.

23. Identity of Cluster 2 needs to be stated earlier in the text when other clusters are described.

24. The authors use terms such as acinar differentiation and acinar specification in terms of different time points. It may be best to first introduce earlier time points E14-E16 (specification) and then later E16-P1 (differentiation).

25. D-G of Figure 4 is validation of acinar specification and differentiation using markers previously described by this group by single cell RNAseq (Hauser et al., 2020). These results do not strongly relate to investigation of FGFR expression and are better suited for a supplementary figure. More appropriately, images should include acinar population markers with co-expression of *Fgfr1* and *Fgfr2* transcripts (FISH) to confirm and validate expression enrichment proposed by single cell RNAseq.

Figure 5

26. The authors describe a significant phenotype of pronounced acinar hypoplasia, reduced gland weight and what appears to be a loss of MIST1 protein expression (Figure 5 D and F) with Aqp5 driven deletion of Fgfr1fl/+;Fgfr2fl/fl and Fgfr1fl/fl;Fgfr2fl/fl, yet gene expression analysis indicates no significant loss in Aqp5, Bhlha15 and Cldn10. This suggests that gene expression analysis may not be the best representation of the cellular signatures in the tissue. This is further emphasized by no significant change in gene expression in Lpo yet images provided indicate an almost complete loss of LPO protein in these samples. It is therefore essential to quantitatively analyze protein expression within these tissues to truly define phenotype and protein expression. This may be done by quantitative methods such as western blot or quantification of immunofluorescent images.

27. The authors use the term “moderately decreased” when describing gene expression of Aqp5, Bhlha15 and Cldn10 while Lpo gene expression appears similar, this gene considered to be defining reduced gene of the Fgfr2b+ population. It is strongly advised to use statistics and terms “significantly” or “non significantly” when describing the increase/decrease of gene expression.

28. The introduction of the SLG to Figure 5 seems misplaced at this stage of the manuscript, when it has not been investigated before. Cell types of the SLG are considered different to the SMG, so it is advised to omit this from the main figure and provide the data in the supplementary figure.

Figure 6:

29. Conclusions made that Fgfr2b works through MAPK for seromucous acinar differentiation have not been supported by experiments and data provided. Firstly, no specific inhibition of Fgfr2b has been carried out, either genetically or pharmacologically. Secondly, the authors use a global inhibition in vitro culture approach, inhibiting multiple pathways and all FGFRs during ex vivo SG development. While their results prove that acinar differentiation is compromised with inhibition of all FGFRs, MAPK and PI3K pathways, receptors and pathways have been inhibited in all cell types of the organ (including mesenchymal, all epithelial cells, neuronal etc). Therefore, no direct link between Fgfr2b signaling, MAPK and seromucous acinar differentiation has been established.

Considering their striking phenotype of seromucous acinar cells in ACID+;Fgfr1fl/fl;Fgfr2fl/fl mice, the authors could analyze these SGs for downstream targets of MAPK signaling (e.g. pMek, pERK). This would provide evidence that MAPK is affected with loss of Fgfr1/Fgfr2 in these cell populations (IF) or globally (western blot). Furthermore, they could culture E15 SMGs with FGFR2b-FC-chimera protein and determine if acinar differentiation is affected (IF staining) and if MAPK signaling is affected (IF/western blot). These experiments will help determine if MAPK is downstream of Fgfr2 driven acinar differentiation.

Figure 7:

30. Results of Figure 7 A are very striking indicating that FGF7 and FGF10 can stimulate differentiation of acinar cells following 24 hours of MAPK inhibition in vitro. To truly define if they are working through MAPK however, it is advised that FGF7 and FGF10 are cultured alongside MAPK inhibitor for the second 24 hours. At this time, the statements made that they are working through MAPK has not been proven. Furthermore, investigation into downstream targets of MAPK signaling with addition of FGF7 and FGF10 would provide further evidence of this cell signaling axis.

31. Inclusion of FGF7 and FGF10 expression in human datasets is interesting however the authors use terminology to suggest FGF7 signaling to acinar cells is directed from MECs, when this has not been proven by any experimental means. FGF7 and 10 are additionally enriched in fibroblasts and ionocytes. Conclusions are made from correlative evidence only and should be removed. Furthermore, human data is in adult glands only. The authors have not shown if Fgfr2 is maintaining acinar identity in the adult, only they have shown it is required during early differentiation processes. While these results are important to include, the role of FGF7 and/or FGF10 in this tissue can only be hypothesized.

Reviewer #2 (Remarks to the Author):

The manuscript by Aure, et al. is built on the solid evidence that FGF signalling is essential for salivary gland morphogenesis and that Fgfr2b is an essential component of the essential pathway. The study uses multiple genetic models, scRNAseq data, in situ hybridisation and immunofluorescent analysis, qPCR and ex vivo culture and pharmacological studies to show that FGFR2b signaling drives seromucous and serous acinar differentiation, via MAPK and FGF7/10.

The conclusions of the manuscript are well validated via the use of multiple genetic knockout models. The re-analysis of an existing scRNAseq dataset is very intuitive and a good demonstration of how the field could and should be utilising existing data. The immunofluorescent staining throughout is beautiful and very indicative of the phenotypes. Furthermore, the loss- and gain-of-function ex vivo studies are really elegant. Overall, the data is robust and well-presented and the paper is very well written, and I just have the following comments or questions that I believe would strengthen the manuscript.

What was the rationale for combining scRNAseq data from P30 and P300? Did the authors check how different expression was before combining, as P30 is still a relatively young age and wouldn't be strictly considered as adult in many studies.

I'm not familiar with the FGF field but is there an Fgfr1a and a Fgfr2a? The authors state that Fgfr1b and Fgfr2b are expressed in the end buds, however, the figure shows expression of Fgfr1 and Fgfr2, with no discerning between different forms. Perhaps this just needs to be made clearer on the figure or in the text or figure legend to avoid confusion?

Could the co-expression of Fgfr1b and Fgfr2b with Krt5 and Bhlha15 be quantified any way? For example, % cells that express only one Fgfr between P1 and adult, versus % cells that express both; and also % cells that express Fgfr(s) and Krt5 or Bhlha15?

Is there any evidence that the use of a constitutive Krt5Cre only deletes expression in basal ductal cells given that it will be "on" throughout development, without the ability to temporally regulate? Given the overlap between Krt5 and Krt14 in the developing/mature salivary gland, is there any risk that a constitutive Krt5Cre deletes throughout the oral epithelium, as the K14Cre does? Having some reassurance here that this is not the case would be really helpful (e.g. previously published data in the salivary gland or fluorescent Cre reporter images). Associated with this and regarding the statement "It also suggests that the duct phenotype observed in the noninducible model was likely due to postnatal differentiation events." – if Krt5 is expressed more extensively throughout the oral epithelium earlier in development, as queried above, this could also be responsible for these differences, surely?

Regarding "Quantification of GFP positive cells showed similar baseline GFP levels in both control and Fgfr1b;Fgfr2b deletion, although expression in males were higher than females (comparison not shown)". Given the known differences in salivary gland structure between male and female mice, personally, I think it would be really interesting to see the comparison data if the authors would consider showing it.

While clearly outside of the scope of this study, have the authors considered trying to delete Fgf7/Fgf10 in MECs (e.g. using a SMA-Cre) to validate their final conclusion that "differentiation of seromucous acinar cells is dependent on FGFR2b-MAPK signaling via FGF7 and FGF10 from MECs as well as fibroblasts (Figure 7H)"? It appears that an Fgf7^{fl/fl} mouse does exist, albeit maybe only very recently (<https://www.biorxiv.org/content/10.1101/2023.02.11.527728v2.full.pdf>).

Response to referees

Reviewer #1 (Remarks to the Author):

Introduction:

1. The introduction is lacking an important explanation of the different stages of gland development, specification, and cellular differentiation, with corresponding timepoints. Without this, it is difficult to understand why the timepoints used at all stages of this study were chosen. We agree with the reviewer and have included a brief summary of gland development and reference papers with more detailed information on the stages of gland development.

The introduction now states on page 4:

“SG development involves similar stages in both humans and mice, and the mouse SMG is a useful model to investigate both early development and cell differentiation^{1, 2, 3}. Mouse SMG development begins at embryonic day (E) 11 as an epithelial invagination into a condensed mesenchyme, and at E12 a primary endbud enlarges and at E13 undergoes branching morphogenesis to give rise to all epithelial cells in the adult gland^{4, 5, 6, 7}. The differentiation of acinar and MECs begins ~E15, so that at birth, postnatal day 1 (P1), functional acinar, MEC and ducts result in salivary secretion^{2, 8}. Further postnatal maturation of acinar, MEC and ductal compartments occurs resulting in functional adult glands.”

2. Background on the need to study salivary gland development and biology as a therapeutic strategy for multiple pathologies is very vague. Further explanation is needed.

We agree and have added some specifics about pathologies and references to reviews on the topic on page 3.

The text now states:

“Loss of acinar cells is a common feature of pathologies including autoimmune diseases such as Sjögren’s disease and a side effect of irradiation therapy for head and neck cancer. Understanding acinar regeneration for the restoration of salivary function continues to be a major clinical challenge^{3, 8, 9}.”

... “This has resulted in an updated view of stemness and plasticity in SGs leading to a renewed focus on niche signals and microenvironments that may inform regenerative therapies¹⁰. Identifying targets that could be used to regenerate salivary function using either molecular, genetic or cell-based approaches is an important therapeutic aim.”

3. The line “Despite the central role of acinar cells,....mucus containing saliva compared with watery serous saliva” is not scientifically sound. Both mucous and serous fluids contain water and mucins. It is suggested to remove this statement and replace with something more appropriate to describe different cell types.

We agree with the reviewer and the text is now edited the text on page 3 as follows:

“Despite the central role of acinar cells, little is known about the developmental mechanisms that drive their specific secretory profiles, although Kras activation was recently shown to promote acinar fate¹¹.”

4. Further introduction to FGFRs and their ligands is required. The authors only discuss Fgfr1b and Fgfr2b, yet both Fgfr1 and Fgfr2 have two major isoforms (IIIb and IIIc). These have not been addressed.

Furthermore, the authors use Fgfr1b and Fgfr2b terminology throughout the manuscript yet their approach using single cell seq and mouse genetics are not isoform specific, therefore they cannot draw conclusions on IIIb isoforms alone.

This is even more important due to previous work defining the Fgfr2c isoform to be essential for SMG development (See Jaskoll et al., 2002). The authors define their results as Fgfr2b specific, yet they cannot rule out the role of Fgfr2c. Terminology should be adjusted to discuss Fgfr1 and Fgfr2 only.

We thank the reviewer for this comment and have updated the text to include more introduction of the FGFRs and isoforms on page 3.

The focus of our investigation is the epithelial FGFRs and although the floxed alleles in our mouse model are not isoform specific by themselves, we are using them in combination with epithelial specific Cre drivers or epithelial cell populations. Since isoforms are expressed in a tissue specific manner (IIIb in epithelial cells) the combination with epithelial specific Cre drivers results in isoform specific deletion. For example, the embryonic deletion of FGFR1 and FGFR2 driven by Krt14Cre does not affect the FGFRc isoform directly since that isoform is exclusively expressed in mesenchymal cells (not expressing activated Cre). Similarly, the scRNAseq itself does not recognize specific isoforms, but we are showing expression in epithelial cell types expressing the epithelial isoforms specifically.

The text on page 3 has been edited as follows:

“Fibroblast growth factor receptors (FGFR) are a family of four receptor tyrosine kinases (FGFR1-4). Alternative splicing of an extracellular Ig-like III domain results in seven functionally distinct receptors expressed in a tissue specific manner; mesenchymal cells express the “c” isoforms, while epithelial cells express the “b” isoforms¹². The epithelial FGFR1b and FGFR2b are activated in an autocrine or paracrine manner by their major ligands FGF7 and FGF10 and require heparan sulfate coreceptors for optimal signaling^{13, 14}.”

We have also included the reference and highlighted the role of Fgfr2c on page 4 as well and the text now reads:

“In addition, the SMGs of FGFR2c^{+/-} hemizygous mice were hypoplastic, highlighting a role for mesenchymal FGF signaling although this is not the focus of this investigation¹⁵.”

5. The statement “Studies using murine models have established that both FGFR1b and FGFR2b are expressed in the epithelium during embryonic development and the ligands, FGF10 and FGF7, are expressed in the mesenchyme” is not referenced.

Thanks for pointing this out, the reference has been added on page 4 and now reads:

“Both FGFR1b and FGFR2b are expressed in the epithelium during murine embryonic development and the ligands, FGF10 and FGF7, are expressed in the mesenchyme¹⁶.”

6. It is difficult to understand the explanation of Fgfr1 and Fgfr2 expression during embryonic timepoints with relation to Figure 1 A. The dot plot is clear, however the text gives a general

overview without discussing individual timepoints shown in the Figure. The authors explain that 88% of cells in the endbud cluster express both *Fgfr1b* and *Fgfr2b* in embryonic glands.

Is this in all timepoints combined?

Percent of expression in dot plots are also heterogenous for both *Fgfr1* and *Fgfr2* yet both receptors are described to have the same expression.

Please review and provide a more detailed analysis of *Fgfr1* and *Fgfr2* expression of each time point during embryonic and postnatal/adult stages.

We agree with the reviewer and have provided additional details to the text. We also caution over interpretation of single cell RNAseq data as only the most highly expressed ~3000 genes are expressed/cell.

The text on page 5 now reads:

“Both *Fgfr1b* and *Fgfr2b* were widely expressed in the embryonic epithelium with generally higher percentage expression at E12 compared to later stages in accordance with our previous reports^{16, 17, 18}. Specifically, *Fgfr1b* expression was detected in endbuds, (88% at E12 to 40% at E16), *Krt19*⁺ ducts (40% at E12 to 20% at E16), and basal ducts (55% at E12 to 40% at E16). *Fgfr2b* was also detected in endbuds (50% at E12 to 20% at E16), *Krt19*⁺ ducts (65% at E12 to 10% at E16), and basal ducts (45% at E12 to 30% at E16) (Fig. 1A). Expression in MECs was detected at E16, with 80% of cells expressing *Fgfr1b* and 40% expressing *Fgfr2b* (Fig. 1A). *Fgfr3b* was detected in only 5-7% of duct cells and *Fgfr4* was barely detectable in E12 endbuds (~1%). Postnatally, *Fgfr1b* and *Fgfr2b* were both expressed in higher percentage of cells at P1, where the gland is undergoing postnatal growth and maturation, compared to adult glands. Specifically, in basal ducts the percentage of cells expressing *Fgfrs* decreased (*Fgfr1b*: 55% at P1 to 16% in adult, *Fgfr2b*: 32% at P1 to 4% in adult). A higher percentage of MECs expressed *Fgfrs* at P1 compared to adult (*Fgfr1b*: 82% at P1 to 31% in adult, *Fgfr2b*: 34% at P1 and 3% in adult). In acinar cells (P1 *Bpifa2*⁺, *Smgc*⁺, *mKi67*⁺ and adult serous or seroumucous) the trends were similar with less percentage of cells expressing *Fgfrs* in adult compared to P1 (*Fgfr1b*: ~26% at P1 to 1% in adult, *Fgfr2b*: 23% at P1 to 5% in adult, Fig. 1B). Also, *Fgfr1b* was detected in 25% of *Gstt1*⁺ intercalated duct cells and *Fgfr2b* was expressed in 20% of P1 ionocytes (Fig. 1B), *Fgfr3b* was detected in 9% of basal duct cells and *Fgfr4b* was detected in <1% of basal ducts and acinar cells.”

7. This group previously defined numerous end bud clusters within these embryonic timepoints, yet one endbud cluster is described here. Please explain if all clusters were collected and analyzed as one and justify the reasoning for this. “Acinar populations previously described” have not been explained.

Although several endbud clusters were described in one of the figures in our prior work, we have here used the endbud annotation which includes all endbud clusters. We are using the same annotation as in the publicly available dataset and endbud refers to all endbud cells for each stage. The further subclustering of endbuds at all embryonic stages did not provide any further information about *Fgfrs*. We also did not want to overinterpret the data, but rather use the general annotations to get an overview of expression patterns.

We have updated the text on page 5 to reflect this:

“The scRNAseq atlas contains several stages of embryonic (E12, E14 and E16) and postnatal development and we followed previous annotations in this dataset.”

8. Figure 1 B shows 3 samples for P1 and 2 adult for acinar cells. This has not been explained. We have now updated Figure 1B to include the specific annotation from the single cell dataset. The figure labels and legends now include *Bpifa2*⁺, *Smgc*⁺ and *mKi67*⁺ as well as seromucous (SM) and serous (S).

The legend now states:

1B) “*Fgfr1b* and *Fgfr2b* were enriched in acinar cells, basal duct, and MECs in postnatal glands. P1: postnatal day 1, Ad: Adult. Subclusters of P1 acinar cells enriched for *Bpifa2*, *Smgc* or *mKi67* and adult acinar cells that are seromucous (SM) and serous (S) are shown.”

9. Authors have defined that they will focus their analysis on *Fgfr1* and *Fgfr2* but it has not been explained why they only investigate transcript expression of P1 and adult in Figure 1 C-D when they have just described embryonic expression.

We thank the reviewer for pointing this out and we have added a clarifying sentence on page 6.

The text now reads:

“We focused our analysis on *Fgfr1b* and *Fgfr2b* due to their higher expression in both embryonic and postnatal development, compared to other *Fgfr* genes. Since embryonic expression patterns have previously been reported^{16,18}, we focused on confirming cell-specific expression patterns in postnatal glands.”

10. Figure 1 B shows *Fgfr1* expression to be more widespread in the *Krt5*⁺ population at P1, yet FISH shows *Fgfr2* transcripts (red) to be a lot more numerous than *Fgfr1* (white) transcripts and *Fgfr1* expression looks very minimal. Please explain and define how many samples were used to validate transcript expression.

We thank the reviewer for this question but caution the direct quantitative comparison of scRNAseq and RNAscope in situ analysis. Our aim with the FISH is to confirm overlap of FGFRs in specific cell types. We have provided examples of these cells that are positive for both FGFRs and *Krt5*. We used 3 different salivary glands for each FISH analysis and show representative example of an area of one gland. This information is now included in the methods.

Figure 1B and 1C are indeed both showing expression of *Fgfr1* and *Fgfr2*, still direct quantitative comparison of expression level between the two methods is not advised. There are technical issues that could account for the differences the reviewer is pointing out. First, the single cell 10x sequencing is polyA tail for all mRNAs while the in situ is probe-based specifically for *Fgfr1* or *Fgfr2* only. Second, the RNA-scope in-situ results are also dependent on the probe hybridization efficiency which may not be identical for each probe, as well as where a particular section is within the cell.

11. The authors vaguely describe “expression pattern was similar to that of adult human SMG, MSG, and PG” yet dot plots provided in supplementary show differing patterns and also the introduction of new cell types that have not previously been defined (e.g. ionocytes). Please describe all data in manuscript that is shown in figures.

For the human data, we have used the annotations provided in the datasets, without introducing new cell types. In the case for the two minor SG datasets, annotations were harmonized for the purpose of integrating them, but no new annotations were made.

We have updated the text on page 6 to include more details of the general expression pattern, and the text now reads as follows:

“We also investigated FGFR expression using scRNAseq datasets from adult human SMG, MSG, and PG^{19, 20, 21}. In general, human glands were similar to adult mouse SMG with expression of FGFR1 and FGFR2 in basal duct (FGFR1: 5-40%, FGFR2: 30-50%) and MECs (FGFR1: 50-70%, FGFR2: 30-40%). Acinar cells also showed expression of FGFR2 (2-60%) and some expression of FGFR1 (1-5%). Similar to mouse SMG, basal cells additionally expressed FGFR3 (11-30% of cells in clusters) and FGFR4 was detected in 2% or less of basal duct cells (Supplementary Fig. 1A).”

Figure 2

12. The authors show beautiful images of phenotypes in both the *Tfap2a-Cre* and *Krt14Cre* mouse lines, yet it has not been explained why two separate Cre drivers have been used. The reasoning and benefits of this need to be described.

We thank the reviewer for this comment. We hypothesized both Cre lines would result in salivary phenotypes, however, since they both are constitutively expressed, differential temporal expression patterns of the two Cre drives in various tissues could affect the outcome when crossed with FGFR floxed mice.

The results show that they indeed have differential degrees of craniofacial and appendage defects. Still, this did not interfere with the results shown in the salivary epithelium and the *Crect-Cre* results also confirmed the ectodermal origin of SMG epithelium, some early literature had suggested it was endoderm derived.

The text on page 7 has been edited as follows:

“We predicted both *Crect* and *Krt14Cre* would result in similar GFP activation within the oral cavity and confirm the ectodermal origin of SG epithelium, although differential temporal expression patterns of the two Cre drivers would result in various defects in other tissues.”

“SGs from the E12 *Crect+;Fgfr1^{fl/fl};Fgfr2^{fl/+}* were comparable to control and formed endbuds with a stratified, invaginating epithelium in the condensing mesenchyme, confirming the ectodermal origin of salivary epithelium (Fig. 2E).”

13. E12 buds look like they are at a younger timepoint in the *Crect-Cre* (Figure 2 E) than the *Krt14Cre* (F) and SOX10 expression has not been confirmed in these samples. It is recommended to stain E12 *CrectCre* samples for SOX10 to determine if acinar progenitor commitment occurs as in the WT samples.

We agree that the *Crect-Cre* presented is early E12 compared to the late E12 images shown in the *Krt14Cre*. We have corrected the labels on the figures to show this. Both Cre lines function to delete FGFR1 and 2 in the salivary epithelium. The *Crect-Cre* also affects craniofacial development. Since both Cre models labels the majority of salivary epithelial cells, indicating widespread FGFR1b/2b deletion, we do not expect acinar progenitor commitment to be different in the two models.

14. Can the authors explain why they believe there is a differing phenotype between these two Cre lines? Single panels of stains are recommended as it is difficult to see SOX10 expression with green and red overlapping.

We have included the single channels in Figure 2 at the reviewer's request.

We do not state that there is a differing phenotype in SG development of these two Cre lines, but rather a difference in craniofacial and limb development. The reason for this could be differential temporal expression patterns of the two Cre drives in various tissues. The *Crect-cre* is ectodermal and is expressed earlier in development although the expression of these 2 cre lines in other appendages is not the focus of the paper, we point out the gross morphological differences in their phenotypes when crossed with *FGFR1/2* floxed lines.

15. It is stated that “Both cre models generated embryos with gross morphological changes that were non-viable due to lack of organ development upon *Fgfr2b* deletion.” Was this confirmed by the authors or assumed based on previous global KO models? It is generally understood that the *Fgfr2b* KO is non-viable due to lung agenesis, yet the lung is endoderm derived.

It would therefore be expected that lung development would not be affected in the *Crect-Cre* and non-viable embryos are caused by cranial abnormalities driven by loss of global *Fgfr2* and further emphasize the importance of *Fgfr2c*.

We thank the reviewer for this comment. The viability issue in the *Crect* model was due to severe craniofacial defects after deletion of *FGFR2*. We have included a sentence highlighting the importance of *Fgfr2c* as well (see point 4). There was no gross phenotype of lung development in the *Crect* mice.

We have updated the text on page 7 as follows:

“While *Crect+;Fgfr1^{fl/+};Fgfr2^{fl/+}* embryos were indistinguishable from *Crect-* controls, *Crect+;Fgfr1^{fl/+};Fgfr2^{fl/fl}* embryos strongly phenocopied other *Fgfr2b* knockout transgenic mice, and were non-viable likely due to severe craniofacial defects upon *Fgfr2b* deletion.”

16. It is not clear why the authors have chosen two separate *Krt5* cre mice. A constitutive *Krt5* Cre is used and only adult female mice are analyzed, and then an inducible *Krt5Cre* is used to delete *Fgfr* alleles during early development. It is interesting that the authors did not use the constitutive cre to investigate embryo development, as results shown in Figure 3 A could be driven by an embryonic or adult phenotype. Please explain.

Initially, we investigated the phenotype in constitutive *Krt5Cre* adult mice because this mouse was available, viable and its skin phenotype had been published²². Based on these results we observed the adult SMGs and hypothesized, as did the reviewer, that the outcome could be driven either from an embryonic or postnatal phenotype. Thus, to follow up on the findings we decided to do temporal deletions and test whether is a pre- or postnatal phenotype, therefore we switched to an inducible system that was available in our lab. This allowed us to test both timepoints with one model. Although we agree that it is possible to do the embryonic

experiments using a constitutive Cre, we decided to not pursue that as we would expect the outcome of those experiments to be similar to what we present in Supplementary Fig. 2.

17. Authors define ductal hypoplasia in glandular tissue sections, yet an increase in the acinar population could also lead to fewer visible ducts with this analysis method. This needs to be investigated further before ductal hypoplasia can be determined. Quantitative gene and/or protein expression analysis of ductal and acinar markers is encouraged to define this.

We thank the reviewer for pointing this out, since the phenotype shows relatively smaller ducts with a reduced area of ducts in sections, which we quantitated, and a relatively larger area of acinar cells, we can't say its fewer ducts or more acinar area, as it is likely a relative reduction in ducts compared to acinar cells.

We have revised the text on page 8 to remove the word hypoplasia as follows:

“The *Krt5Cre+;Fgfr^{fl/fl};Fgfr2^{fl/fl}* SGs had a distinct duct phenotype with 60% reduction duct/total gland area and loss of granular convoluted tubules (GCT), which are sexually dimorphic, specialized ducts producing NGF and EGF in mice (Fig. 3B and 3C).”

Figure legend 3C has been edited as follows:

“Quantification of duct/gland ratio showed significant reduction in duct area after *Fgfr1b* and *Fgfr2b* deletion in *Krt5+* lineage in adult SMGs. Unpaired t-test was used to calculate significance (* $p < 0.05$).”

18. It appears by the images provided in Figure 3 that male mice have a significant ductal phenotype with 90 day chase of inducible *Krt5* driven deletion of *Fgfr1* and *Fgfr2*. Mis-organization of ductal cells and collapse of the luminal network seem apparent by ECAD and nuclear staining, yet the authors have not commented on this. Do the authors believe FGFRs have a more significant role in male GCT homeostasis vs female? Single cell analysis in adult mice was carried out on females only, so it cannot be confirmed if FGFRs have the same cellular expression in males as females. FISH of *Fgfr1* and *Fgfr2* in adult male mice is advised to support the inclusion of male glands in this figure.

We thank the reviewer for the comment, and we agree that the male ducts do appear to have an additional phenotype that may not be directly explained by decreased lineage tracing. The female glands also have a similar phenotype, although less pronounced.

We do not have direct evidence to conclude whether FGFRs are more significant for male GCTs compared to females. Postnatal development of GCTs is androgen dependent and the reviewer makes an interesting proposition that could warrant further investigation, still, we think this is beyond the scope of this study. Also, since human salivary glands do not have GCTs we were not interested in following this line of investigation.

The single cell data from P30 indeed includes both male and female glands. However, very few basal cells were identified within the P30 dataset making it challenging to make strong conclusions about FGFR expression levels in males vs females. As the reviewers suggests, we have included in situ from adult male glands which confirmed similar expression overlap between *Fgfr1/2* and *Krt5* as well as *Bhlha15*. This is now included in Figure 1C and D.

We have added the following to the text on page 9 addressing the duct phenotype:

“Additionally, non GFP+ duct cells exhibited abnormal morphology after a 90-day chase in *Krt5rtTA;tetCre+;Fgfr1^{fl/fl};Fgfr2^{fl/fl};mTmG* glands, suggesting disruption of duct integrity or homeostasis.”

19. Lineage tracing is very informative of the role of Fgfrs in progenitor cells replenishment of the GCT cells. The quantification method used to calculate GFP+ cells is questionable, as different protein content of cells can contribute to different GFP cell membrane intensity (or intracellular intensity as observed in male GCTs). GFP+ cells as a percentage of all GCT cells should be calculated for each condition to determine lineage of Krt5 progenitors.

It is well established that lineage tracing from basal cells to GCTs and other luminal duct cells occurs during homeostasis and has been done by several groups using various Cre drivers including Krt5^{5, 7, 23, 24, 25}.

Previous quantification by others have shown a significant increase in labeled duct cells after 3 months. We therefore chose the 3-month timepoint and we were indeed able to detect significant lineage tracing comparable to previous reports in our control glands using the current quantification method. We therefore conclude that the method used is adequate and a valid method to compare the control to the *Krt5rtTA;tetCre;Fgfr1^{fl/fl};Fgfr2^{fl/fl}* glands.

20. Figure 3 D schematic is confusing, as it outlines the hypothesis and not a result. Schematics should be used to define the result of an experiment so it is recommend to adjust this and place at the end of Figure 3 to confirm discovery.

We agree and have revised the schematic and moved it to the end of the figure.

The text on page 9 has been revised and now reads:

“These results show that FGFR1b and FGFR2b signaling is required for basal progenitor contribution to duct homeostasis in adult male and female SMGs (Fig. 3G).”

Figure 4

21. The reason for combining E16 and P1 datasets to analyze acinar cells needs to be more clearly described (e.g. previously defined stages of acinar differentiation).

We agree, and the text on page 10 has been edited to state this more clearly:

“To further analyze FGFR expression at previously defined stages of acinar differentiation, we bioinformatically isolated and re-clustered E16 and P1 acinar cells (Fig. 4A and Supplementary Fig. 3A).”

22. The authors highlight a population of cells that were eliminated based on ambiguous expression of markers of multiple cell lineages, however the group of genes listed are known molecular signatures of myoepithelial cells and may indicate differentiating MECs from end bud cells. This population may hold significance due to high FGFR expression in the MEC population (as described earlier in Figure 1) and should be further investigated and/or exclusion needs to be further justified. Please see Chatzeli et al., 2023 *Developmental Cell* with regards to early MEC differentiation.

We thank the reviewer for this point and resource. Although cluster 4 may be interesting, we are focusing on acinar differentiation and detailed investigation of the function of FGFRs in potential myoepithelial lineages is beyond the scope of this paper. Still, we did not detect lineage tracing to MECs in our ACID Cre model and there were no significant changes in MEC markers

in any of the ACID;flox genotypes, suggesting that cluster 4 is either not a major contributor to the MEC lineage, it is compensated for by other sources, or the cluster is simply remaining doublets in the scRNAseq data.

The text on pages 10 has been updated to clarify the elimination of the cluster and reads as follows:

“Cluster 4 expressed relatively low levels of the canonical acinar markers and had additional genes from multiple cell lineages that do not align with acinar cell identity including but not limited to *Krt14*, *Krt5*, *Trp63*, *Acta2*, *Colla1* and *Col3a1*. Although doublets were previously removed from the dataset, it is not clear whether this small cluster are remaining doublets or cells in a transitional cell state¹¹. Due to this ambiguity, and to focus specifically on acinar cells, this cluster was excluded from further analysis.”

23. Identity of Cluster 2 needs to be stated earlier in the text when other clusters are described. The text has been edited to mention the identity of Cluster 2 earlier.

The text on page 10 and 11 now states:

“Based on average expression, cluster 2 (Fig. 4C) appeared to be double positive for *Fgfr1b* and *Fgfr2b*; however, this cluster contained either *Smgc*⁺ or *Bpifa2*⁺ cells, and was defined by proliferative markers such as *Mki67*, *Bub1b*, *Aurka* and *Top2a* (Supplementary Fig. 3C). Pathway analysis indicated active proliferation as the major functional state of these cells (Supplementary Fig. 3C).”

...”Additionally, proliferating acinar cells were found within both subpopulations (SMGC and LPO), further suggesting that cluster 2 is mainly proliferating cells made up by a mix of the two distinct populations rather than a unique population (Fig. 4I).”

24. The authors use terms such as acinar differentiation and acinar specification in terms of different time points. It may be best to first introduce earlier time points E14-E16 (specification) and then later E16-P1 (differentiation).

We are referring to specification and differentiation as two events. One being the specification of acinar cell type identified as CLDN10, MIST1 and AQP5 expression and one being detection of the subpopulations (*Smgc*/*Fgfr1* and *Bpifa2*/*Fgfr2*).

The text on page 10 is modified to highlight this as follows:

“To establish the timeline for acinar specification (onset of canonical acinar markers) and differentiation (specific acinar subpopulations), we performed qPCR of embryonic SMGs at E14, E15 and E16, showing detection of acinar specification by E14 and differentiation of subpopulations by E15 (Fig. 4D).”

25. D-G of Figure 4 is validation of acinar specification and differentiation using markers previously described by this group by single cell RNAseq (Hauser et al., 2020). These results do not strongly relate to investigation of FGFR expression and are better suited for a supplementary figure. More appropriately, images should include acinar population markers with co-expression of *Fgfr1* and *Fgfr2* transcripts (FISH) to confirm and validate expression enrichment proposed by single cell RNAseq.

The authors disagree with the reviewer, we think that the use of acinar differentiation markers that are expressed in the acinar subpopulations does relate to the FGFR story and is important evidence that the cre-deletion of FGFRs in these cells results in reduction in the proteins produced by these specific subpopulations. We are establishing the timeline of marker expression, confirming, and establishing localization patterns of genes we identified in the scRNAseq and putting them into the context of *Fgfr1* and *Fgfr2* enriched populations. These markers and patterns are important to establish clearly since they are used as markers in later experiments/figures. We already have included FISH showing that the *Fgfr1* and *Fgfr2* transcripts are expressed in acinar cells, after Cre-deletion the expression of the FGFR proteins may not be detectable.

Figure 5

26. The authors describe a significant phenotype of pronounced acinar hypoplasia, reduced gland weight and what appears to be a loss of MIST1 protein expression (Figure 5 D and F) with *Aqp5* driven deletion of *Fgfr1*^{fl/+};*Fgfr2*^{fl/fl} and *Fgfr1*^{fl/fl};*Fgfr2*^{fl/fl}, yet gene expression analysis indicates no significant loss in *Aqp5*, *Bhlha15* and *Cldn10*.

This suggests that gene expression analysis may not be the best representation of the cellular signatures in the tissue.

This is further emphasized by no significant change in gene expression in *Lpo* yet images provided indicate an almost complete loss of LPO protein in these samples.

It is therefore essential to quantitatively analyze protein expression within these tissues to truly define phenotype and protein expression. This may be done by quantitative methods such as western blot or quantification of immunofluorescent images.

We agree with the reviewer and have included quantification of protein staining. This shows significant loss of acinar cells after FGFR2b deletion. To show changes in protein expression within acini, we quantified acinar cells in all genotypes. This confirmed the significant reduction in LPO and MUC10, but not SMGC and the results are now included in Figure 5.

The text on pages 11-12 is edited as follows:

... “Still, gene expression of *Aqp5*, *Bhlha15* and *Cldn10* were detected in all genotypes (Fig. 5C). In line with this, AQP5, MIST1 and CLDN10 expressing cells were found in all genotypes, although acinar organization was severely disrupted and MIST1⁺ cells per area was significantly reduced after *Fgfr2b* deletion (Fig. 5D and 5H). After *Fgfr2b* deletion, there was no change in the defining genes for *Fgfr1b*⁺ cells (*Smgc*, *Gstt1*, or *Ramp1*), while expression of *Bpifa2*, *Proll1*, *Mucl2*, and *Elf5* was significantly reduced with a similar trend for *Lpo* (Fig. 5C). Interestingly, *Bpifa2*, *Proll1*, and *Mucl2* were also decreased in *Fgfr1*^{fl/+};*Fgfr2*^{fl/+} and *Fgfr1*^{fl/fl};*Fgfr2*^{fl/+}. This was due to heterozygous *Fgfr2b* rather than *Fgfr1b* deletion as *Fgfr1*^{+/+};*Fgfr2*^{fl/+}, gave similar results (data not shown). Protein detection of the two populations showed loss of LPO and MUC10 after *Fgfr2b* deletion (Fig. 5E, 5F). Furthermore, all acinar cells (MIST1⁺) expressed SMGC after *Fgfr2b* deletion, indicating a complete loss of seromucous differentiation (Fig. 5G). When focusing on areas with acini, a significant decrease in LPO and MUC10 but not SMGC was quantified after *Fgfr2b* deletion (Fig. 5I).”

Figure legend 5H):

“Quantification of IHC staining showed a significant decrease of MIST1+ cells per area after *Fgfr2b* deletion (n=3). One-way ANOVA with Dunnett’s test for multiple comparisons to WT control (*p =0.005).”

Figure legend 5I):

“Quantification of protein staining showed significant decrease in LPO and MUC10, while SMGC did not significantly change following *Fgfr2b* deletion (n=3 for each stain). One-way ANOVA with Dunnett’s test for multiple comparisons to WT control (*p =0.005).”

27. The authors use the term “moderately decreased” when describing gene expression of *Aqp5*, *Bhlha15* and *Cldn10* while *Lpo* gene expression appears similar, this gene considered to be defining reduced gene of the *Fgfr2b*+ population. It is strongly advised to use statistics and terms “significantly” or “non significantly” when describing the increase/decrease of gene expression. We thank the reviewer for pointing this out and we have updated the text to reflect this. Please see text edits under point 26.

28. The introduction of the SLG to Figure 5 seems misplaced at this stage of the manuscript, when it has not been investigated before. Cell types of the SLG are considered different to the SMG, so it is advised to omit this from the main figure and provide the data in the supplementary figure.

We agree and the data is now moved to Supplementary Figure 4.

Figure 6:

29. Conclusions made that *Fgfr2b* works through MAPK for seromucous acinar differentiation have not been supported by experiments and data provided. Firstly, no specific inhibition of *Fgfr2b* has been carried out, either genetically or pharmacologically.

Secondly, the authors use a global inhibition in vitro culture approach, inhibiting multiple pathways and all FGFRs during ex vivo SG development.

While their results prove that acinar differentiation is compromised with inhibition of all FGFRs, MAPK and PI3K pathways, receptors and pathways have been inhibited in all cell types of the organ (including mesenchymal, all epithelial cells, neuronal etc). Therefore, no direct link between *Fgfr2b* signaling, MAPK and seromucous acinar differentiation has been established. Considering their striking phenotype of seromucous acinar cells in ACID+;*Fgfr1*^{fl/fl};*Fgfr1*^{fl/fl} mice, the authors could analyze these SGs for downstream targets of MAPK signaling (e.g. pMek, pERK).

This would provide evidence that MAPK is affected with loss of *Fgfr1*/*Fgfr2* in these cell populations (IF) or globally (western blot).

Furthermore, they could culture E15 SMGs with FGFR2b-FC-chimera protein and determine if acinar differentiation is affected (IF staining) and if MAPK signaling is affected (IF/western blot). These experiments will help determine if MAPK is downstream of *Fgfr2* driven acinar differentiation.

We thank the reviewer for this comment and experimental suggestions. Firstly, using the ACID model, deletion of *Fgfr2* occurs specifically in AQP5+ acinar cells which only express the epithelial isoform, *Fgfr2b*. This is therefore in effect *Fgfr2b* specific and shows a clear and specific connection between *Fgfr2b* and seromucous and serous acinar differentiation.

At the same time, we recognize and agree with the reviewer that the ex vivo organ culture is testing inhibitors globally. We also agree that culturing E15 SMGs with FGFR2b-FC-chimera protein, measure acinar differentiation and connect it to decreased MAPK signaling is a good way to further strengthen our conclusion.

We have included these experiments and results in Figure 6 and Supplementary Figure 5 and edited the text on page 13 as follows:

“Expression of *Cdh1*, *Fgfr1b*, *Fgfr2b*, *Fgf10*, *Fgf7*, *Smgc*, *Tesc* and *Lman1* was not affected by any of the inhibitors tested, except rFGFR2b increased *Fgf10* expression (Supplementary Fig. 5F and Fig. 6C). Expression of the FGFR downstream effectors, *Etv4* and *Etv5*, was reduced after rFGFR2b, pan-FGFR, and MAPK inhibitor treatment, confirming inhibition of the FGFR signaling pathway (Supplementary Fig. 5F). There was a trend of reduced expression of *Bhlha15*, *Cldn10*, and *Aqp5* expression after pan-FGFR and MAPK inhibitor treatments. Although, *Aqp5* expression was significantly reduced after MAPK inhibition, indicating a specific MAPK signaling dependence for *Aqp5* and *Cldn10* was significantly decreased after FGFR inhibition (Fig. 6C). Both *Lpo* and *Bpifa2* expression was decreased after rFGFR2b, pan-FGFR, MAPK and PI3K inhibitor treatment, whereas PLC γ inhibitor treatment did not affect *Fgfr2b*-dependent gene expression (Fig. 6C). In addition, SGs treated with rFGFR2b showed decreased phosphorylated ERK, indicating the gene expression changes detected after rFGFR2b treatment were due to decreased MAPK pathway activity (Supplementary Fig. 5G). Gross histology of all treatment groups was comparable to control after 24hr treatment (Supplementary Fig. 5H) and luminal AQP5, nuclear MIST1 and lateral CLDN10 were detected in all groups indicating that acinar specification was not affected (Fig. 6D). Staining for SMGC showed expression of the protein in all groups, while LPO was not detected after pan-FGFR or MAPK inhibitor treatment (Fig. 6E). Taken together, these results show that differentiation of seromucous acinar cells requires FGFR2b signaling through the MAPK pathway.”

Figure legends have been added as follows:

Figure 6C) “Gene expression of canonical acinar markers were decreased after FGFR inhibition. *Cldn10* was decreased after pan-FGFR inhibitor (SU5402, 5 μ g/ml). *Aqp5* and *Lpo* were decreased after MAPK inhibitor (UO126, 20 μ g/ml), while *Bpifa2* was decreased after rFgfr2b (20 μ g/ml), pan-FGFR- and MAPK-inhibitors. Genes enriched in the *Fgfr1b*⁺ population did not change with the various inhibitors compared to vehicle control. One-way ANOVA with Dunnett’s test for multiple comparisons to control and students t-test when comparing two groups (*p<0.05).”

Supplementary Figure 5:

F) Receptors, ligands and downstream effectors in E15 organ culture after 24 hrs of inhibitor treatments compared to their vehicle control. One-way ANOVA with Dunnett’s test for multiple comparisons to control and students t-test when comparing two groups (rFGFR2b), *p<0.05.

G) E15 SMGs cultured for 24 hrs with rFGFR2b showed reduction in pERK/tERK compared to vehicle control. Graphs shows quantification of Western Blot (n=3), images are representative bands from the two groups.

Figure 7:

30. Results of Figure 7 A are very striking indicating that FGF7 and FGF10 can stimulate differentiation of acinar cells following 24 hours of MAPK inhibition in vitro. To truly define if they are working through MAPK however, it is advised that FGF7 and FGF10 are cultured alongside MAPK inhibitor for the second 24 hours. At this time, the statements made that they are working through MAPK has not been proven. Furthermore, investigation into downstream targets of MAPK signaling with addition of FGF7 and FGF10 would provide further evidence of this cell signaling axis.

We thank the reviewer for this suggestion and have added the experiments showing no significant increase in *Aqp5*, *Bpifa2* or *Lpo* following MAPK inhibitor and FGF7 or FGF10 for the second 24 hrs and these results have been added to Figure 7B. We have also measured downstream target pERK and show comparable levels between control and FGF7 or FGF10 treatment as opposed to significant decrease after MAPK inhibitor or washout alone and this is included in Figure 7C.

The text on page 14 has been edited to include these results:

“Addition of FGF7 or FGF10 for 24 hrs along with continued MAPK inhibitor treatment showed no changes in *Aqp5*, *Bpifa2*, or *Lpo* gene expression compared to inhibitor alone, confirming that the FGF7 or FGF10 stimulated changes act through the MAPK pathway (Fig. 7B). Accordingly, phosphorylated ERK was decreased following MAPK inhibitor and washout, while FGF7 or FGF10 treatment was comparable to control confirming FGFR2b-dependent MAPK activation (Fig. 7C).”

Figure legends for Figure 7 have been edited as follows:

B) Genes reduced following MAPK inhibition could be partially rescued by inhibitor washout. *Aqp5* and *Lpo* were further increased by FGF7 and FGF10 treatment (n=4). FGF7 or FGF10 had no effect in the presence of MAPK inhibitor (n=3). One-way Anova with Dunnett’s test for multiple comparisons (**p<0.0001, *p<0.05).

C) Western Blot showed decrease in pERK/tERK after MAPK inhibitor treatment and washout. Addition of FGF7 or FGF10 restored pERK/tERK comparable to control. One-way Anova with Dunnett’s test for multiple comparisons *p<0.05).

31. Inclusion of FGF7 and FGF10 expression in human datasets is interesting however the authors use terminology to suggest FGF7 signaling to acinar cells is directed from MECs, when this has not been proven by any experimental means. FGF7 and 10 are additionally enriched in fibroblasts and ionocytes. Conclusions are made from correlative evidence only and should be removed. Furthermore, human data is in adult glands only. The authors have not shown if *Fgfr2* is maintaining acinar identity in the adult, only they have shown it is required during early differentiation processes. While these results are important to include, the role of FGF7 and/or FGF10 in this tissue can only be hypothesized.

We agree with the reviewer and the text on page 15 has been edited as follows:

“In vivo, MECs directly contact and wrap around the acini, and this acinar complex is surrounded by the basement membrane, whereas the stromal fibroblasts that produce FGF10 are separated from acini by the basement membrane. Taken together, we hypothesize that FGF7 secreted from MECs that are in direct contact to acini provide critical niche signal for

seromucous acinar differentiation. Taken together, we have shown that seromucous acinar cell differentiation is dependent on FGFR2b-MAPK signaling via FGF7 and FGF10 (Fig. 7H).”

Reviewer #2 (Remarks to the Author):

1. What was the rationale for combining scRNAseq data from P30 and P300? Did the authors check how different expression was before combining, as P30 is still a relatively young age and wouldn't be strictly considered as adult in many studies.

We thank the reviewer for raising this question. We initially analyzed the two adult stages separately and found the overall expression pattern to be similar and the two stages were combined to increase the cell numbers for each cluster. While P30 is a relatively young adult, comparisons between young and old adults could give further insight to FGFR signaling and expression patterns during aging. However, some clusters, such as basal cells and MECs, are not well represented in the separate adult stages and we aimed to identify which cell types express FGFRs in postnatal glands in general to be guided to potential cell types to focus on further.

We have clarified this in the text on page 5 and it now reads:

“There were no major differences in P30 and P300, and these stages were combined to increase cell numbers and referred to as adult in our analysis.”

2. I'm not familiar with the FGF field but is there an *Fgfr1a* and a *Fgfr2a*? The authors state that *Fgfr1b* and *Fgfr2b* are expressed in the end buds, however, the figure shows expression of *Fgfr1* and *Fgfr2*, with no discerning between different forms. Perhaps this just needs to be made clearer on the figure or in the text or figure legend to avoid confusion?

We agree that more information about FGFR splice b and c splice isoforms and FGF ligands is needed, and we have included more this in the introduction (in response to point 4 from reviewer 1) and made text and figure legends consistent in terms of epithelial FGFR1b and FGFR2b.

3. Could the co-expression of *Fgfr1b* and *Fgfr2b* with *Krt5* and *Bhlha15* be quantified any way? For example, % cells that express only one *Fgfr* between P1 and adult, versus % cells that express both; and also % cells that express *Fgfr(s)* and *Krt5* or *Bhlha15*?

We thank the reviewer for this question. It is challenging to draw strong conclusions about the specific percentages in different cell populations with the RNAscope technique we have used here and do not want to overinterpret these results. Also, scRNAseq technique highlights cells that have high expression of a gene as usually only the most highly expressed 3000 genes are detected, so expression by dot plots should not be over interpreted but taken in context with the FISH data. Our point was to show cell specific overlap with both methods to help guide us to specific cell populations.

We cannot exclude that there is overlap with *Fgfr1* and *Fgfr2* P1 acinar cells and we realize that our wording might have been easily misunderstood. There is enrichment in different cells, although not exclusive.

We have modified the text to highlight this on page 6:

“There was also widespread *Fgfr1b* and *Fgfr2b* coexpression with the acinar marker *Bhlha15*⁺ (MIST1, Fig. 1D). Notably, at P1, acinar cells were enriched for either *Fgfr1b* or *Fgfr2b*, although not exclusive, while adult acinar cells were overall enriched for both receptors (Fig. 1D).”

4. Is there any evidence that the use of a constitutive Krt5Cre only deletes expression in basal ductal cells given that it will be “on” throughout development, without the ability to temporally regulate? Given the overlap between Krt5 and Krt14 in the developing/mature salivary gland, is there any risk that a constitutive Krt5Cre deletes throughout the oral epithelium, as the K14Cre does?

Having some reassurance here that this is not the case would be really helpful (e.g. previously published data in the salivary gland or fluorescent Cre reporter images). Associated with this and regarding the statement “It also suggests that the duct phenotype observed in the noninducible model was likely due to postnatal differentiation events.” – if Krt5 is expressed more extensively throughout the oral epithelium earlier in development, as queried above, this could also be responsible for these differences, surely?

We agree that constitutive Krt5Cre will be “on” throughout development without temporal regulation. Although Krt5 and Krt14 overlap in postnatal glands, during early development, Krt5 expression is restricted in the oral epithelium whereas Krt14 is more widely expressed.

The fact glands developed in the constitutive Krt5Cre model, suggests the restricted expression pattern of Krt5 in the oral epithelium.

If the Krt5-Cre was activated widely in the oral epithelium like Krt14, which gives rise to the entire glands, we would expect a phenotype with no glands similar to the Krt14-Cre and Cre models. No significant differences in duct development detected at birth, further suggests that the observed phenotype is due to events happening after embryonic development.

We have added references addressing the expression patterns of both Sox10 and Krt5 early in gland development on page 8:

“Next, we aimed to determine whether the expression of FGFRs in basal ducts is required for their development. At E12, *Sox10* expressing endbud cells give rise to the gland parenchyma, while KRT5 has a more restricted expression pattern in the oral epithelium and SG duct as previously reported^{4, 26}.”

We also have modified the sentence in the end of the paragraph (on page 8) to highlight that the phenotype observed is most likely due to events occurring after gland initiation:

...” These results points to a specific role for FGFR signaling in basal progenitors either during duct development or postnatal GCT differentiation.”

5. Regarding “Quantification of GFP positive cells showed similar baseline GFP levels in both control and *Fgfr1b;Fgfr2b* deletion, although expression in males were higher than females (comparison not shown)”. Given the known differences in salivary gland structure between male and female mice, personally, I think it would be really interesting to see the comparison data if the authors would consider showing it.

We thank the reviewer for this suggestion and have included the comparison in Supplementary Figure 2I.

The text on page 9 is edited to reflect this:

“Quantification of GFP expression showed similar baseline levels in both control and with *Fgfr1b;Fgfr2b* deletion, although expression in males was higher than females (Fig. 3F and Supplementary Fig. 2I).”

Figure S2I legend is added as follows:

“Quantification of GFP expression in IHC sections at day 0 showed similar levels of expression in control and after *Fgfr1b* and *Fgfr2b* deletion although lower detection was observed in females compared to male glands.”

5. While clearly outside of the scope of this study, have the authors considered trying to delete *Fgf7/Fgf10* in MECs (e.g. using a *aSMA-Cre*) to validate their final conclusion that “differentiation of seromucous acinar cells is dependent on FGFR2b-MAPK signaling via FGF7 and FGF10 from MECs as well as fibroblasts (Figure 7H)”? It appears that an *Fgf7^{fl/fl}* mouse does exist, albeit maybe only very recently (<https://www.biorxiv.org/content/10.1101/2023.02.11.527728v2.full.pdf>).

We thank the reviewer for this resource and agree that this is outside the scope of this paper but would be a very interesting and important experiment to further validate our final working model.

We have included the following in the discussion on page 17 to highlight this:

“Whether such treatment increases the seromucous and serous acinar cells remains to be determined, and further work including deletion of *Fgf7* in MECs using a recently reported *FGF7^{fl^{ox27}}* may support our working model for the critical role of niche cell signals.”

1. de Paula F, Teshima THN, Hsieh R, Souza MM, Nico MMS, Lourenco SV. Overview of Human Salivary Glands: Highlights of Morphology and Developing Processes. *Anat Rec (Hoboken)* **300**, 1180-1188 (2017).
2. Tucker AS. Salivary gland development. *Seminars in cell & developmental biology* **18**, 237-244 (2007).
3. Patel VN, Hoffman MP. Salivary gland development: a template for regeneration. *Seminars in cell & developmental biology* **25-26**, 52-60 (2014).
4. Athwal HK, et al. Sox10 Regulates Plasticity of Epithelial Progenitors toward Secretory Units of Exocrine Glands. *Stem Cell Reports* **12**, 366-380 (2019).
5. Song EAC, et al. Genetic and scRNA-seq Analysis Reveals Distinct Cell Populations that Contribute to Salivary Gland Development and Maintenance. *Scientific Reports* **8**, (2018).
6. Emmerson E, et al. SOX2 regulates acinar cell development in the salivary gland. *Elife* **6**, (2017).

7. May AJ, *et al.* Diverse progenitor cells preserve salivary gland ductal architecture after radiation-induced damage. *Development* **145**, 12 (2018).
8. Chibly AM, Aure MH, Patel VN, Hoffman MP. Salivary gland function, development, and regeneration. *Physiol Rev* **102**, 1495-1552 (2022).
9. Rocchi C, Emmerson E. Mouth-Watering Results: Clinical Need, Current Approaches, and Future Directions for Salivary Gland Regeneration. *Trends Mol Med* **26**, 649-669 (2020).
10. Rocchi C, Barazzuol L, Coppes RP. The evolving definition of salivary gland stem cells. *NPJ Regen Med* **6**, 4 (2021).
11. Chatzeli L, *et al.* A cellular hierarchy of Notch and Kras signaling controls cell fate specification in the developing mouse salivary gland. *Dev Cell* **58**, 94-109.e106 (2023).
12. Ornitz DM, Itoh N. The Fibroblast Growth Factor signaling pathway. *Wiley Interdisciplinary Reviews-Developmental Biology* **4**, 215-266 (2015).
13. Ornitz DM, Itoh N. New developments in the biology of fibroblast growth factors. *WIREs Mech Dis* **14**, e1549 (2022).
14. Xie Y, *et al.* FGF/FGFR signaling in health and disease. *Signal Transduct Target Ther* **5**, 181 (2020).
15. Jaskoll T, *et al.* Embryonic submandibular gland morphogenesis: stage-specific protein localization of FGFs, BMPs, Pax6 and Pax9 in normal mice and abnormal SMG phenotypes in FgfR2-IIIc(+/ Δ), BMP7(-/-) and Pax6(-/-) mice. *Cells, tissues, organs* **170**, 83-98 (2002).
16. Patel VN, Rebutini IT, Hoffman MP. Salivary gland branching morphogenesis. *Differentiation* **74**, 349-364 (2006).
17. Steinberg Z, *et al.* FGFR2b signaling regulates ex vivo submandibular gland epithelial cell proliferation and branching morphogenesis. *Development* **132**, 1223-1234 (2005).
18. Patel VN, *et al.* Specific heparan sulfate structures modulate FGF10-mediated submandibular gland epithelial morphogenesis and differentiation. *J Biol Chem* **283**, 9308-9317 (2008).
19. Costa-da-Silva AC, *et al.* Salivary ZG16B expression loss follows exocrine gland dysfunction related to oral chronic graft-versus-host disease. *iScience* **25**, 103592 (2022).
20. Huang N, *et al.* SARS-CoV-2 infection of the oral cavity and saliva. *Nat Med* **27**, 892-903 (2021).

21. Jones RC, *et al.* The Tabula Sapiens: A multiple-organ, single-cell transcriptomic atlas of humans. *Science* **376**, eabl4896 (2022).
22. Ramirez A, *et al.* A keratin K5Cre transgenic line appropriate for tissue-specific or generalized Cre-mediated recombination. *Genesis* **39**, 52-57 (2004).
23. Weng PL, Aure MH, Maruyama T, Ovitt CE. Limited Regeneration of Adult Salivary Glands after Severe Injury Involves Cellular Plasticity. *Cell Rep* **24**, 1464-1470.e1463 (2018).
24. Kwak M, Alston N, Ghazizadeh S. Identification of Stem Cells in the Secretory Complex of Salivary Glands. *J Dent Res* **95**, 776-783 (2016).
25. Kwak M, Ghazizadeh S. Analysis of histone H2BGFP retention in mouse submandibular gland reveals actively dividing stem cell populations. *Stem Cells Dev* **24**, 565-574 (2015).
26. Chatzeli L, Gaete M, Tucker AS. Fgf10 and Sox9 are essential for the establishment of distal progenitor cells during mouse salivary gland development. *Development* **144**, 2294-2305 (2017).
27. Manieri E, *et al.* Defining the structure, signals, and cellular elements of the gastric mesenchymal niche. *bioRxiv*, (2023).

REVIEWER COMMENTS

Reviewer #1 (Remarks to the Author):

I thank the authors for addressing my original comments. It appears misunderstandings have been made. I have addressed my continual concerns below. Please review original comments and new comments. (Numbers match to original comments submitted)

4. I thank the authors for clarifying their justification of the continual use of terms Fgfr2b and Fgfr1b. I understand that their deletion strategy is epithelial specific so it can be assumed that they are targeting these epithelial isoforms. The use of the terms in specific areas to describe their results however are based on this assumption alone and have not been biologically confirmed and therefore are misleading. Their genetic tools (scRNAseq and FISH) show results of complete Fgfr1 and Fgfr2 expression, as clarified by the authors in their figures with labelling of Fgfr1 and Fgfr2 only (not isoform specific). They therefore can only describe the results as such in the Figure titles, legends and manuscript (e.g. title of Figure 1 is “The epithelial isoforms, Fgfr1b and Fgfr2b, are broadly expressed in mouse SMG”, when no panel in the figure confirms this, only Fgfr1 and Fgfr2). As these are the tools used, the authors can only describe the results with terms Fgfr1 and Fgfr2 (The authors have done this in the figure panels already and even confirm they practice caution in over interpretation of the scRNAseq data). These changes are required throughout the entire manuscript. I believe this does not take away from the study, as it is simply the scientific sound way of describing the genetic tools used and the appropriate results.

5. New reference added describing isoform and ligand expression is not accurate. These expression patterns have been described much earlier than Wells et al., 2013. FGF work on the SG from this lab even precedes this study.

6. Please remove “our” previous reports. These referenced studies are not from this group. Thank you for describing in more detail. Discussion E14 in these results also seems important as percentage seems quite high as E12, especially Fgfr1 and Fgfr2 in end bud cells.

7. Please provide reference to previous annotations in newly added sentence.

10. The justification described needs to be included in the manuscript. Highlight importance of overlapping expression, but transcript number may be different.

11. The authors seem to have misunderstood my comment. New annotations have not been included compared to the original datasets published, but they are new in terms of this manuscript (i.e. ionocytes have not been described or introduced leading up to this point). The inclusion of this cell type is sudden and unexplained.

13. The authors have ignored the recommended review experiment and have not explained sufficiently as to why. They have mentioned they do not expect any differences in the two models but the phenotype is different in the images provided.

14. The images provided indicate a different phenotype with the early *Crect-Cre* driven deletion giving rise to an elongated bud and the *Krt14Cre* driven deletion, albeit older, appears as an arrested early E11 bud. Cellular organization is also different. Please review original comment.

16. Thank you for this explanation. Please include in the main manuscript so the rationale is clear.

17. The authors have not given sufficient evidence still that a ductal phenotype occurs and have only changed the phrasing slightly, still defining a ductal phenotype. Please review original comment and experimental recommendations.

18. I thank the authors for including expression in adult male mice. As per my original comment, I advise adding a description for the differing phenotype between male and female in the main manuscript.

19. The authors have misunderstood my original comment. The method of quantification, GFP intensity is questionable and GFP+ cells or all GCT cells is advised.

21. The authors have not understood the request and justification as to why E16 and P1 were clustered needs to be explained.

25. There seems to be a misunderstanding here. To clarify, the establishment of gene and protein expression is relevant to the FGFR story, but the data within this figure from D-G is not relevant to FGFRs themselves. Furthermore, the information that compiles most of this figure (D-G) is not new information to the field. Expression of these genes and proteins over developmental time have been published previously in the SMG (for some examples see Larsen et al., 2011, Das et al., 2009, Nielsen et al, 2013, May et al., 2022). As it stands in the main figure, it appears this is new information. This is why addition as a supplementary figure is more appropriate. The reader can correlate protein expression in terms of FGFR expression, but the data is not being advertised as new discoveries.

26. The quantification of protein staining has provided more informative information to Figure 5. The authors have kept gene expression data, which does not match up with protein expression (see original comment) yet this has not been discussed or explained. This needs to be addressed further.

29. I appreciate the inclusion of some new experiments based of my suggestion, but I am still concerned that not enough evidence is provided to back up statements made linking FGFR2b to acinar differentiation via MAPK. Global gene expression patterns have been studied and short WB analysis included in the supplementary does not seem significant to conclude statements (also not acinar specific). As mentioned before, further analysis (IF and WB) of the ACID+Fgfrfl/fl;Fgfrfl/fl mouse and IF and WB of culture conditions would help to confirm this. The authors have not addressed why these were not included. Proliferation and cell death analysis also needs to be included for this new condition.

Reviewer #2 (Remarks to the Author):

The authors have adequately addressed all of my comments and I have no further questions or concerns. Thank you.

RESPONSE TO REFEREES

Reviewer #1 (Remarks to the Author):

4. I thank the authors for clarifying their justification of the continual use of terms *Fgfr2b* and *Fgfr1b*. I understand that their deletion strategy is epithelial specific so it can be assumed that they are targeting these epithelial isoforms.

The use of the terms in specific areas to describe their results however are based on this assumption alone and have not been biologically confirmed and therefore are misleading. Their genetic tools (scRNAseq and FISH) show results of complete *Fgfr1* and *Fgfr2* expression, as clarified by the authors in their figures with labelling of *Fgfr1* and *Fgfr2* only (not isoform specific). They therefore can only describe the results as such in the Figure titles, legends and manuscript (e.g. title of Figure 1 is “The epithelial isoforms, *Fgfr1b* and *Fgfr2b*, are broadly expressed in mouse SMG”, when no panel in the figure confirms this, only *Fgfr1* and *Fgfr2*). As these are the tools used, the authors can only describe the results with terms *Fgfr1* and *Fgfr2* (The authors have done this in the figure panels already and even confirm they practice caution in over interpretation of the scRNAseq data). These changes are required throughout the entire manuscript. I believe this does not take away from the study, as it is simply the scientific sound way of describing the genetic tools used and the appropriate results.

We agree with the reviewer and have removed references to the “b” splice isoforms and have revised the text describing scRNAseq and FISH analysis as suggested, and we have highlighted the targeting of epithelial isoforms when using epithelial specific Cres and floxed mice. Where we have used isoform specific tools, such as isoform-specific primers for qPCR and recombinant FGFR2b, we refer to the splice isoforms.

Example from page13:

“This showed consistent expression of *Fgfr1b* and *Fgfr2b* using isoform specific PCR primers, *Fgf10* and *Fgf7*, and the downstream effectors *Etv4*, and *Etv5* (Supplementary Fig. 5B).”

5. New reference added describing isoform and ligand expression is not accurate. These expression patterns have been described much earlier than Wells et al., 2013. FGF work on the SG from this lab even precedes this study.

The original reviewer comment was to add a reference to the sentence, which we did.

The actual reference added is Patel et al 2006 not Wells et al 2013.

The sentence was added on page 4 and reads:

“Both FGFR1b and FGFR2b are expressed in the epithelium during murine embryonic development and the ligands, FGF10 and FGF7, are expressed in the mesenchyme¹⁴.”

This reference is: Patel VN, Rebutini IT, Hoffman MP. Salivary gland branching morphogenesis. *Differentiation* 74, 349-364 (2006).

The reason for this confusion is that the reference number provided in the “response to referees” document is 16 and not 14. This is because the response document had its own reference list (so the ref number is not identical to what is in the main manuscript document). We apologize that this was not clear. We do, however, think this is an appropriate reference.

6. Please remove “our” previous reports. These referenced studies are not from this group. Thank you for describing in more detail. Discussion E14 in these results also seems important as percentage seems quite high as E12, especially *Fgfr1* and *Fgfr2* in end bud cells.

We suspect the same confusion with reference numbers has happened here as above. The references given in the main manuscript as well as comments to referees are indeed from this group and are:

Patel VN, Rebutini IT, Hoffman MP. Salivary gland branching morphogenesis. *Differentiation* **74**, 349-364 (2006).

Steinberg Z, *et al.* FGFR2b signaling regulates ex vivo submandibular gland epithelial cell proliferation and branching morphogenesis. *Development* **132**, 1223-1234 (2005).

Patel VN, *et al.* Specific heparan sulfate structures modulate FGF10-mediated submandibular gland epithelial morphogenesis and differentiation. *J Biol Chem* **283**, 9308-9317 (2008).

We have added the actual % for E14 from the dot plots in Fig 1A into the results as requested. The text on page 5 now reads:

“Both *Fgfr1* and *Fgfr2* were widely expressed in the embryonic epithelium with generally higher percentage expression at E12 compared to later stages in accordance with our previous reports^{14, 34, 35}. Specifically, *Fgfr1* expression was detected in endbuds, (88% at E12, 78% at E14 to 40% at E16), Krt19+ ducts (40% at E12 and E14 to 20% at E16), and basal ducts (55% at E12 to 40% at E16). *Fgfr2* was also detected in endbuds (50% at E12, 40% at E14 to 20% at E16), Krt19+ ducts (65% at E12, 20% at E14 to 10% at E16), and basal ducts (45% at E12 to 30% at E16) (Fig.1A). Expression in MECs was detected at E16, with 80% of cells expressing *Fgfr1* and 40% expressing *Fgfr2* (Fig. 1A).”

7. Please provide reference to previous annotations in newly added sentence.

We have added the reference to the newly added sentence:

Hauser BR, Aure MH, Kelly MC, Hoffman MP, Chibly AM, Genomics Computational Biol C. Generation of a Single-Cell RNAseq Atlas of Murine Salivary Gland Development. *iScience* **23**, 35 (2020).

The text on page 5 now reads:

“The scRNAseq atlas contains several stages of embryonic (E12, E14 and E16) and postnatal development and we followed previous annotations in this dataset³⁰.”

10. The justification described needs to be included in the manuscript. Highlight importance of overlapping expression, but transcript number may be different.

We have added text to highlight that we used in RNAscope in situ hybridization to show overlap with known cell markers on page 6 and agree that the actual transcript number of each FGFR may be different in individual cells of a specific cell type.

The text on page 6 reads:

“We focused our analysis on *Fgfr1* and *Fgfr2* due to their higher expression in both embryonic and postnatal development, compared to other *Fgfr* genes. Since embryonic expression patterns

have previously been reported^{14, 35}, we focused on confirming cell-specific expression patterns in postnatal glands. RNAscope in situ hybridization was used to highlight the overlap of FGFR expression with known cell types in postnatal glands, although the actual transcript number of each FGFR in individual cells of a specific cell type may be different.”

11. The authors seem to have misunderstood my comment. New annotations have not been included compared to the original datasets published, but they are new in terms of this manuscript (i.e. ionocytes have not been described or introduced leading up to this point). The inclusion of this cell type is sudden and unexplained.

We thank the reviewer for clarifying this comment and we have included a mention of ionocytes earlier in the paper on page 3 as follows, although they are not the focus of this study:

“In vivo, acinar cells are surrounded by myoepithelial cells (MEC) that wrap around and directly contact them, and are connected to duct cells, containing basal cells and ionocytes, that modify and transport saliva into the oral cavity.”

13. The authors have ignored the recommended review experiment and have not explained sufficiently as to why. They have mentioned they do not expect any differences in the two models but the phenotype is different in the images provided.

We have added the SOX10 staining into the E12 images as initially asked by the reviewer.

We did not observe a difference in salivary gland phenotype with either the *Crect* or *K14-cre*, and we do not have evidence to suggest any specific phenotype in salivary glands between the two Cre lines. The images provided are of different early developmental stages, which may attribute to what the reviewer refers to as a difference, although we state in the text that overall our analysis suggests the salivary glands with both Cres are similar. The control and *CRECT* samples at E12 do not have detectable SOX10 expression in the epithelium.

The outcome of crossing any of the Cre and floxed lines depends on two events, the Cre activation and FGFR deletion genotypes. Considering the identical Cre activation in the salivary epithelium observed in both *Krt14Cre* and *Crect*, we would not expect them to be different because any phenotype would then be attributed to the FGFR deletion (and not the Cre activation). We have no reason to think that identical FGFR deletion in the same cell pool would lead to different outcomes and acinar commitment at this point. We therefore do not consider it appropriate to state differences in salivary glands between these Cre lines in the manuscript.

14. The images provided indicate a different phenotype with the early *Crect*-Cre driven deletion giving rise to an elongated bud and the *Krt14Cre* driven deletion, albeit older, appears as an arrested early E11 bud. Cellular organization is also different. Please review original comment.

The original comment asked us to explain differences, we cannot explain a differing salivary gland phenotype as reviewer is suggesting because we do not have evidence supporting this. Again, the images provided are of developmental stages ~0.5-1 day difference, which attribute to the appearance of a difference. Images of early developmental stages may also differ from section to section depending on the angle of the cut and where in the gland the cut is made, and we do not want to make statements about cellular organization. Further studies, beyond the scope of this investigation, would be required to determine if there are differences in the reduction of both FGFRs in the cre-expressing cells, due to differences in the levels of FGFR deletion with each of the two different cre-drivers. Additionally, it would also require a more in-depth analysis

of the three-dimensional structure and cell numbers to investigate such a presumed potential phenotype, which is beyond the scope of this paper. Please see the above comment as to why we do not hypothesize that there would be a difference in bud formation or acinar commitment between the two Cre lines.

16. Thank you for this explanation. Please include in the main manuscript so the rationale is clear. We have added the following statements on page 8 in the manuscript to highlight the reason for using the inducible Cre:

“The *Krt5Cre+;Fgfr^{1/1};Fgfr^{2/1}* SGs had a distinct duct phenotype with 60% reduction duct/total gland area and loss of granular convoluted tubules (GCT), which are sexually dimorphic, specialized ducts producing NGF and EGF in mice (Fig. 3B and 3C). These results point to a specific role for FGFR signaling in basal progenitors either during duct development or postnatal GCT differentiation.

To further dissect the temporal role of FGFRs in duct lineages, we used the inducible *Krt5rtTA;tetCre* mice crossed with *Fgfr^{1/1};Fgfr^{2/1}* and mTmG strains to investigate whether the observed phenotype in the non-inducible model is due to pre or postnatal events.”

17. The authors have not given sufficient evidence still that a ductal phenotype occurs and have only changed the phrasing slightly, still defining a ductal phenotype. Please review original comment and experimental recommendations.

In the original comment the reviewer suggested that we did not have sufficient data to define duct hypoplasia without adding further analysis. We agreed with this assessment and did indeed change the wording. However, the term “phenotype” is referring to observable physical properties determined by genotype, including measurable characteristics, and we think there is sufficient evidence to say there is a ductal phenotype, without further experimental evidence. The gland in the *Krt5Cre FGFR1/2 fl/fl* is smaller, it weighs ~40% less (Suppl Fig 2E). In order to quantify if the reduction in gland size is due to a smaller acinar or ductal compartment (Fig 3C) we measured the area of ducts in sections and measure a significantly smaller relative duct area compared with the rest of the acinar area after FGFR deletion. While this does not rule out an acinar phenotype, it shows that a greater proportion of the reduction in gland size is due to the reduction in the ductal compartment. We also followed up these experiments by performing time specific lineage tracing which showed that duct progenitor function in postnatal glands is dependent of FGFR signaling, further supporting a ductal phenotype. We therefore think there is sufficient evidence to state that we observe a duct phenotype.

We have reworded the text to highlight the reduced size of the *Krt5-cre* salivary gland: the text on page 8 states:

“In addition, the SGs were significantly smaller (~40%) in adult *Krt5Cre+;Fgfr^{1/1};Fgfr^{2/1}* mice compared to wildtype (WT, *Krt5Cre-*, Supplementary Fig. 2E).”

18. I thank the authors for including expression in adult male mice. As per my original comment, I advise adding a description for the differing phenotype between male and female in the main manuscript.

We don't want to over-speculate on the comparisons of male and female duct phenotype evident from the lineage tracing. But at the insistence of the reviewer, we have added a brief comment on

the phenotype of male and female glands. Previously the reviewer asked us to comment “Do the authors believe FGFRs have a more significant role in male GCT homeostasis vs female?” Our data suggest there is a reduction in GCTs with the K5-cre and although there is a nonsignificant reduction in GFP+ intensity in the male compared with the female (Fig 3F), we can speculate that FGFRs may play a role in GCT homeostasis. GCTs are a prominent cell feature of male murine SGs.

We can only comment of the GFP positive cells that have undergone Cre activation and subsequent FGFR1/2 deletion, therefore, any phenotype observed in non-GFP expressing cells can only be indirectly linked to the FGFR deletion itself. It is difficult to speculate on any specific mechanism that may lead to a duct phenotype using the genetic experimental setup we used.

We have edited the text on page 9 and suggest that this warrants further work:

“Additionally, non GFP+ duct cells exhibited abnormal morphology after a 90-day chase in *Krt5rtTA;tetCre+;Fgfr1^{fl/fl};Fgfr2^{fl/fl};mTmG* glands, suggesting a disruption of duct integrity or homeostasis, or alternatively loss of a paracrine factor secreted by FGFR expressing duct cells, although further work is needed to confirm this. Quantification of GFP expression showed similar baseline levels in both control and with *Fgfr1;Fgfr2* deletion, although expression in males was higher than females (Fig. 3F and Supplementary Fig. 2I) likely due to the prominence of GCT ducts in male murine SGs and potentially suggesting FGFRs may have a role in GCT homeostasis.”

And on page10:

“These results show that FGFR1 and FGFR2 signaling is required for basal progenitor contribution to duct homeostasis in adult male and female SMGs (Fig. 3G).”

19. The authors have misunderstood my original comment. The method of quantification, GFP intensity is questionable and GFP+ cells or all GCT cells is advised.

The reviewer is asking us to change the normalization of our analysis from nuclear staining to GCT staining. However, the GFP is in the ductal compartment that is made up of more than just the GCTs and the authors disagree with the reviewer’s assessment.

Our controls are in agreement with the well-known previously published outcome of Krt5 basal duct lineage tracing with the timepoints used. Previous quantifications by others have used various models all coming to similar conclusions. This suggests that there is more robustness to this experiment than the reviewer is considering. Since we measure significant contribution (ie GFP+ cells) in our controls, similar to others, we conclude that the measurements and method are adequate. We therefore do not think the quantitation is questionable and do not agree that changing the method for quantification is necessary.

21. The authors have not understood the request and justification as to why E16 and P1 were clustered needs to be explained.

The original comment asked us to describe more clearly the reason for combining E16 and P1 datasets to analyze acinar cells. We agreed with the reviewer’s assessment, and it is because these stages include previously defined stages of acinar differentiation. We have now added one more sentence to clarify why this is useful.

The text on page 10 now reads:

“To further analyze FGFR expression at previously defined stages of acinar differentiation, we bioinformatically isolated and re-clustered E16 and P1 acinar cells (Fig. 4A and Supplementary Fig. 3A). This allowed detection of cell specific genes during the establishment and differentiation of acinar sub populations.”

25. There seems to be a misunderstanding here. To clarify, the establishment of gene and protein expression is relevant to the FGFR story, but the data within this figure from D-G is not relevant to FGFRs themselves. Furthermore, the information that compiles most of this figure (D-G) is not new information to the field. Expression of these genes and proteins over developmental time have been published previously in the SMG (for some examples see Larsen et al., 2011, Das et al., 2009, Nielsen et al, 2013, May et al., 2022). As it stands in the main figure, it appears this is new information. This is why addition as a supplementary figure is more appropriate. The reader can correlate protein expression in terms of FGFR expression, but the data is not being advertised as new discoveries.

We are pleased that the reviewer agrees establishing gene and protein expression is relevant to the FGFR story. Again, we disagree with the reviewer that most of the staining in this figure should be supplementary, as it shows the reader very clearly the development of the two acinar subpopulations with known acinar markers.

We are well aware of previous publications the reviewer mentions. However, these papers have shown expression of some of these markers, but the novelty of the figure is that we have shown the markers in combination in a developmental context as the acinar subpopulation differentiation occurs. We acknowledge the papers showing the established acinar specific markers (AQP5, MIST1 and CLDN10), but the combination of staining with these markers have not been shown. This is directly relevant to the results we present in the paper as we use these markers to show how they are affected by FGFR deletion. The images are central for the readers to interpret the rest of the manuscript. We therefore do not agree that the data should be put in supplementary figures.

We have, however, clarified the text so the reader knows these are all previously established markers of specific cell types.

The text on page 10 and 11 now reads:

“To verify the timeline for acinar specification (onset of canonical acinar markers) together with differentiation (specific acinar subpopulations) using previously established markers and genes we identified in specific *Fgfr* enriched populations, we performed qPCR of embryonic SMGs at E14, E15 and E16, showing detection of acinar specification by E14 and differentiation of both subpopulations by E15 (Fig. 4D). Immunostaining showed detection at E15 with a progressively increasing organization of luminal AQP5, basolateral CLAUDIN10 (CLDN10) and nuclear MIST1 (Fig. 4E). Markers for the two subpopulations were detected from E16, and by late E16 both populations were clearly distinguished with immunostaining by SMGC and LPO proteins (Fig. 4F). In P1 glands, the two acinar populations were visualized through immunostaining with either SMGC and LPO or GSTT1 and MUC10 (Fig. 4G, H). Co-staining of SMGC and GSTT1 or LPO and MUC10 further confirmed overlapping expression of these markers (Supplementary Fig. 3D, E). Additionally, proliferating acinar cells were found within both subpopulations (SMGC and LPO), further suggesting that cluster 2 is mainly proliferating cells made up by a mix of the two distinct populations rather than a unique population (Fig. 4I).

Taken together, *Fgfr1* and *Fgfr2* were enriched within specific acinar subpopulations that could be identified by specific markers and highlights E16 and P1 as appropriate stages to investigate the establishment of both subpopulations. We hypothesized that FGFR signaling would differentially affect expression of defining markers and either the development or maturation of these acinar subpopulations.”

26. The quantification of protein staining has provided more informative information to Figure 5. The authors have kept gene expression data, which does not match up with protein expression (see original comment) yet this has not been discussed or explained. This needs to be addressed further.

The original comment and requested was to add quantification of protein expression, which we provided and further strengthened our conclusions. We also know that gene expression levels and changes do not necessarily match changes in protein expression. Transcription, translation and protein stability are all regulated differently. The gene expression data stands and just because it might not all match up with protein staining levels is no reason to leave it out. Indeed, the data does match for canonical acinar markers, which were detected both by qPCR and immunostaining in all genotypes. For the subpopulation specific markers, we detect sustained expression of FGFR1b enriched markers and reduced *Fgfr2b* enriched markers using both qPCR and IHC. Differing results for *Lpo* gene expression and protein may be due to post-transcriptional regulation or protein turnover.

We edited the text on page 12:

“ Furthermore, all acinar cells (MIST1+) expressed SMGC after *Fgfr2* deletion, highlighting the complete loss of seromucous differentiation (Fig. 5G). When focusing on seromucous acinar proteins, quantification showed a significant decrease in LPO, while gene expression was still detected suggesting post-transcriptional regulation (Fig 5I). Similarly, MUC10 (*Proll*) staining was decreased after *Fgfr2* deletion, consistent with its gene expression (Fig. 5I).

29. I appreciate the inclusion of some new experiments based of my suggestion, but I am still concerned that not enough evidence is provided to back up statements made linking FGFR2b to acinar differentiation via MAPK.

Global gene expression patterns have been studied and short WB analysis included in the supplementary does not seem significant to conclude statements (also not acinar specific). As mentioned before, further analysis (IF and WB) of the ACID+*Fgfr1/fl*;*Fgfr1/fl* mouse and IF and WB of culture conditions would help to confirm this. The authors have not addressed why these were not included. Proliferation and cell death analysis also needs to be included for this new condition.

We have now included IF results of the new condition (recombinant *Fgfr2b*) as suggested by the reviewer which shows changes in cell death and proliferation as well as decrease of *Fgfr2* dependent acinar marker LPO.

We do understand the suggestion to do further analysis (IF and WB) in the ACID ACID+*Fgfr1/fl*;*Fgfr1/fl* mouse model and of culture conditions, however, there are some major experimental caveats that need to be pointed out that make this unfeasible.

The main issue is that we are losing the cell population (Fgfr2b dependent acinar cells) and their specific markers in both the ACID mouse model and organ culture after deletion or inhibition of Fgfr2b or MAPK.

Performing IF to measure decreases in pERK may seem like a straightforward experiment, however, this requires analyzing pERK in a subpopulation that is barely detectable or not there, or is a minor population compared to the rest of the gland (ie the Fgfr2b enriched acinar cells after FGFR2b deletion/inhibition). In other words, it is not possible to specifically measure decrease of downstream targets (pERK) in a cell population that cannot be detected or that is not present. This dilemma is the same in both model systems we used.

We have included WB analysis after Fgfr2b inhibition, which showed decreased pERK although not significant, suggesting decreased MAPK signaling, which is what we would expect. We agree with the reviewer that there are caveats with WB of pERK being a global analysis and not acinar specific. Other pathways signaling via MAPK are present and may also affect the outcome. However, we performed gene expression analysis and IF of acinar markers showing very specific targeting of genes in agreement with MAPK inhibition. While the gene expression analysis is also global, the genes in question are indeed acinar cell specific.

Performing WB on ACID samples brings up the same challenge as for organ culture. Again, we do not detect the FGFR2b enriched cells in the 2fl/fl genotypes, so WB measurement and interpretation would be challenging and questionable. Generating enough samples of the lethal genotypes to perform WB on ACID samples to test this is a major undertaking especially considering the caveats outlined here.

For these reasons we did not include IF staining suggested and focused on the downstream analysis that was possible to do. We apologize if we were not clear about these concerns in our previous response. The loss of function experiment should also be taken into context of the gain of function experiments which does show increased pERK with FGF stimulation together with rescue of acinar markers.

REVIEWERS' COMMENTS

Reviewer #1 (Remarks to the Author):

The authors have made suitable adjustments to meet my concerns. I would like to thank them for their time and consideration of my review and comments.